# Preoperative Inflammatory Markers and the Risk of Postoperative Delirium in Patients Undergoing Lumbar Spinal Fusion Surgery

**DOI:** 10.3390/jcm11144085

**Published:** 2022-07-14

**Authors:** Jin Seo Yang, Jae Jun Lee, Young-Suk Kwon, Jong-Ho Kim, Jong-Hee Sohn

**Affiliations:** 1Department of Neurosurgery, Chuncheon Sacred Heart Hospital, Hallym University College of Medicine, Chuncheon-si 24253, Korea; yang@hallym.or.kr; 2Department of Anesthesiology and Pain Medicine, Chuncheon Sacred Heart Hospital, Hallym University College of Medicine, Chuncheon-si 24253, Korea; iloveu59@hallym.or.kr (J.J.L.); gettys@hallym.or.kr (Y.-S.K.); poik99@hallym.or.kr (J.-H.K.); 3Institute of New Frontier Research Team, Hallym University College of Medicine, Chuncheon-si 24252, Korea; 4Department of Neurology, Chuncheon Sacred Heart Hospital, Hallym University College of Medicine, Chuncheon-si 24253, Korea

**Keywords:** postoperative delirium, spine surgery, neutrophil-to-lymphocyte ratio, monocyte-to-lymphocyte ratio, CRP-to-albumin ratio

## Abstract

We investigated the possible associations between postoperative delirium (POD) and routinely available preoperative inflammatory markers in patients undergoing lumbar spinal fusion surgery (LSFS) to explore the role of neuroinflammation and oxidative stress as risk factors for POD. We analyzed 11 years’ worth of data from the Smart Clinical Data Warehouse. We evaluated whether preoperative inflammatory markers, such as the neutrophil-to-lymphocyte ratio (NLR), the monocyte-to-lymphocyte ratio (MLR), and the CRP-to-albumin ratio (CAR), affected the development of POD in patients undergoing LSFS. Of the 3081 subjects included, 187 (7.4%) developed POD. A significant increase in NLR, MLR, and CAR levels was observed in POD patients (*p* < 0.001). A multivariate analysis showed that the second, third, and highest quartiles of the NLR were significantly associated with the development of POD (adjusted OR (95% CI): 2.28 (1.25–4.16], 2.48 (1.3–4.73], and 2.88 (1.39–5.96], respectively). A receiver operating characteristic curve analysis showed that the discriminative ability of the NLR, MLR, and CAR for predicting POD was low, but almost acceptable (AUC (95% CI): 0.60 (0.56–0.64], 0.61 (0.57–0.65], and 0.63 (0.59–0.67], respectively, *p* < 0.001). Increases in preoperative inflammatory markers, particularly the NLR, were associated with the development of POD, suggesting that a proinflammatory state is a potential pathophysiological mechanism of POD.

## 1. Introduction

Delirium is a common postoperative complication. Postoperative delirium (POD) following various operative procedures has a reported incidence of 10–77% [1,2,3]. POD usually results in adverse outcomes, such as functional disability, an increased length of hospital stay, increased health care costs, and higher morbidity and mortality rates [4,5].

Thus, a better understanding of the incidence and risk factors of POD may help reduce these problems and their associated costs. However, the incidence of POD differs widely, and the reported risk factors differ among studies. Several hypotheses have been proposed to explain POD, such as neuroinflammation, neuro-aging, neuroendocrine stress, neurotransmitter dysregulation, oxidative stress, sleep/wake dysregulation, and network-dysconnectivity. Most of these hypotheses are complementary, rather than internally competitive, with many overlaps and reciprocal influences [6]. Inflammation and oxidative stress may both be involved in the pathophysiology of POD.

The markers of inflammation and oxidative stress are utilized in research and clinical practice to diagnose and monitor inflammation and oxidative stress. Previous reports indicate that delirium is associated with increases in immune-inflammatory biomarkers, including interleukin (IL)-1, IL-6, soluble tumor necrosis factor (TNF) receptor, and other cytokines [7,8]. Additionally, meta-analyses have reported that POD is correlated with the concentrations of peripheral and cerebrospinal fluid (CSF) inflammatory markers, including C-reactive protein (CRP), IL-6, -8, and -10, and TNF-α [9,10]. These studies showed that certain inflammatory markers are associated with POD.

However, their use in clinical practice is limited due to due to cost or a cumbersome diagnostic procedure. The differential white blood cell (WBC) count is routinely obtained in the majority of hospitalized patients, at no additional cost. The neutrophil-to-lymphocyte ratio (NLR) and platelet-to-lymphocyte ratio (PLR) based on the differential WBC count are readily available markers of generalized inflammation, as reported previously [11,12]. These peripheral inflammatory markers are better predictors of mortality and clinical outcome in various medical conditions than the traditional infection markers, including CRP or the total leukocyte count [13,14].

Some studies have shown that peripheral inflammatory markers, such as the NLR and PLR, are associated with the development of POD after hip surgery, cardiac surgery, and esophagectomy [15,16,17,18]. Other studies have reported that the association between the preoperative blood levels of inflammatory mediators and POD may be affected by the type of surgery [9,10]. The special circumstances associated with certain types of surgery may contribute to the etiology of POD.

With the increasing number of older patients undergoing spinal surgery, the rate of POD is expected to increase [19]. However, no study has investigated whether preoperative peripheral inflammatory markers are associated with the development of POD after spine surgery. The relationship between blood inflammatory markers and POD remains controversial. We hypothesized that inflammation and oxidative stress are important antecedents of POD in patients undergoing spine surgery. Therefore, we investigated the possible associations between POD and routinely available preoperative inflammatory markers in patients undergoing lumbar spinal fusion surgery, to explore the utility of inflammation and oxidative stress as risk factors.

## 2. Materials and Methods

### 2.1. Study Population

We retrospectively analyzed data from the Smart clinical data warehouse (CDW) of Hallym University Medical Center (HUMC) from January 2011 to September 2021. The Smart CDW, which is based on the QlikView Elite Solution Provider (Qlik Technologies Inc., Radnor, PA, USA), is used at the five hospitals of the HUMC and provides analysis of electronic medical record text data, as well as integrated analysis of “fixed” data. The study subjects were patients aged ≥18 years who underwent lumbar spinal fusion surgery under general anesthesia during the 11-year study period (1 January 2011–30 September 2021). Exclusion criteria included the development of delirium before surgery, unconsciousness before surgery, undergoing another surgery simultaneously, undergoing sedation after surgery, undergoing ventilator therapy after surgery, and missing medical records. This study was approved by the Clinical Research Ethics Committee of Chuncheon Sacred Heart Hospital, Hallym University (IRB No. 2021-09-005). Since only de-identified data were used in this study, the review board waived the requirement for informed consent.

### 2.2. Data Collection and Parameters

POD patients were defined as those who received postoperative psychiatric counseling and they had shown specific symptoms and signs of delirium, according to the consultation notes [20]. POD was assessed daily by a nurse using the short-form Korean Nursing Delirium Screening Scale [21]. When a nurse suspected POD, the patient received psychiatric counseling and the POD diagnosis was confirmed. We created a list of words indicating specific symptoms that was based on electronic medical records (Appendix A). We collected the medical records, including nursing notes and request notes for consultation with a psychiatrist within 48 h after surgery. When the psychiatrist responded to the request for consultation, POD was confirmed, and the participant was included in the POD group. Other neuropsychiatric patients with symptoms similar to delirium could be excluded on the basis of the list.

We collected the results of preoperative peripheral blood tests within 1 week before surgery to investigate preoperative inflammatory markers, including the complete blood cell count, CRP, and albumin levels. The NLR was calculated by dividing the number of neutrophils by the number of lymphocytes, the MLR was determined by dividing the number of monocytes by the number of lymphocytes, and the PLR was calculated by dividing the number of platelets by the number of lymphocytes. Additionally, the CAR was calculated by dividing the CRP level by the albumin level.

Other perioperative variables were adjusted to prevent bias. The covariates included the patient’s general characteristics, preoperative comorbidities and medications, and characteristics of the anesthetics and surgery. The preoperative variables were age; gender; body mass index; American Society of Anesthesiology physical status > 2; emergency surgery; preoperative comorbidities such as hypertension (HTN), diabetes, heart disease, stroke, cancer, Parkinson’s disease, dementia, depression, kidney disease, liver disease, insomnia, and sleep disorder; medications used before surgery (including calcium channel blockers, beta-blockers, angiotensin-converting enzyme inhibitors, angiotensin-receptor blockers, anti-depressants, hypnotics, antipsychotics, non-steroidal anti-inflammatory drugs, and analgesics, and excepting non-steroidal anti-inflammatory drugs, muscle relaxants, steroids, anti-platelet agents, anti-coagulants, anti-hyperlipidemic drugs, antihistamines, H2-blockers, and miscellaneous); alcohol use; and smoking, We created a list of drugs (Appendix A).

The intraoperative variables were fluid balance, “surgical range” of the lumbar spinal fusion surgery, and operation time. The surgical range was defined according to the number of vertebrae involved in the surgery. Postoperative variables included postoperative intensive care unit care and patient-controlled analgesia. We included the definitions of the variables as a Appendix A.

### 2.3. Statistical Analyses

Continuous data are presented as medians and interquartile ranges, and categorical data as frequencies and percentages. The Mann–Whitney test was performed to compare the continuous data of patients with and without POD. Categorical data were analyzed using the chi-square test. Inflammatory markers were categorized into four groups by quartiles. We calculated the fully adjusted ORs and 95% confidence intervals (CIs) for developing POD using multivariate logistic regression to evaluate the association between the quartiles of the inflammatory markers, among other variables, and the development of POD. The receiver operating characteristic (ROC) curve analysis was performed to determine the optimal cutoffs. The Youden index was used to calculate the cutoff values of the inflammatory markers. All *p*-values were two-sided and a *p*-value < 0.05 was considered significant. SPSS software (version 26.0; IBM Corp., Armonk, NY, USA) was used for statistical analyses.

## 3. Results

A total of 3081 patients underwent lumbar spinal fusion surgery under general anesthesia at one of the five hospitals of the HUMC from January 2011 to September 2021. After excluding 563 patients who met at least one exclusion criterion, 2518 patients were initially included in the study. We excluded 56 patients who developed delirium before surgery, 30 who were unconscious before surgery, 27 who underwent another surgery simultaneously, 18 who underwent sedation therapy after surgery, 102 who underwent ventilator therapy after surgery, and 339 who had missing data in their medical records. Of the enrolled patients, 342 had POD according to the nursing records, and 187 cases were confirmed by a psychiatrist. Thus, the incidence of POD in our study was 7.4%. The flow chart of the enrollment process is provided in Figure 1.

We analyzed the preoperative, intraoperative, and postoperative risk factors of the POD and non-POD groups (Table 1). In the comparison of the preoperative inflammatory markers between the POD and non-POD groups, the NLR (2.68 [1.90, 4.19] vs. 2.17 [1.54, 3.29], *p* < 0.001), MLR (0.29 [0.21, 0.39] vs. 0.23 [0.17, 0.33], *p* < 0.001), and CAR (0.47 [0.25, 1.89] vs. 0.26 [0.22, 0.67], *p* < 0.001) were higher in the POD than the non-POD group (Table 2). The unadjusted odds ratios for the effect of inflammatory markers on developing POD are summarized in Table 3. Additionally, the adjusted ORs of the inflammatory markers and other variables for developing POD are summarized in Table 4. A multivariate logistic regression analysis showed that the risk factors for POD were as follows: Parkinson’s disease (OR 4.46, 95% CI 1.76–11.29; *p* = 0.002), postoperative intensive care unit care (OR 3.83, 95% CI 2.59–5.65; *p* < 0.001), dementia (OR 3.8, 95% CI 1.4–10.31; *p* = 0.009), anti-psychotic drug use (OR 3.52, 95% CI 2.08–5.97; *p* < 0.001), and depression (OR 2.64, 95% CI 1.14–6.16; *p* = 0.024). The second, third, and highest quartiles of NLR were significantly associated with the development of POD (OR = 2.26, 95% CI: 1.24–4.14; OR = 2.47, 95% CI: 1.3–4.72; OR = 2.86, 95% CI: 1.38–5.92, respectively) (Table 3). The ROC curve analysis showed that the discriminative ability of the NLR, MLR, and CAR for predicting POD was low, but almost acceptable (area under the curve [AUC] = 0.60, 95% CI: 0.56–0.64; AUC = 0.61, 95% CI: 0.57–0.65; AUC = 0.63, 95% CI: 0.59–0.67, respectively, *p* < 0.001) (Figure 2). The ROC curves revealed the following cutoff values for the preoperative inflammatory markers to discriminate between the POD and non-POD groups: NLR, 2.26; MLR, 0.26; CAR, 0.38 (Table 5).

## 4. Discussion

This study examined the associations between preoperative blood inflammatory markers and POD in patients undergoing lumbar spinal fusion surgery. We enrolled patients who had undergone lumbar spinal fusion surgery at the HUMC over 11 years using the Smart CDW. Of the 3081 enrolled patients, 187 developed POD. A significant increase in the NLR, MLR, and CAR was observed in the POD group (*p* < 0.001). A multivariate analysis showed that the second, third, and highest quartiles of the NLR were significantly associated with the development of POD. The ROC curve revealed that the discriminative ability of the NLR, MLR, and CAR for predicting POD development was low, but almost acceptable.

In our study, the prevalence of POD after lumbar spinal fusion surgery was 7.4%. The prevalence of POD after spinal surgery ranged from 3.8–40.4% and 0.84–27.6% in two previous meta-analyses [22,23]. The prevalence of POD in our study was relatively low compared to previous studies. The retrospective assessment of POD might have led to an underestimation of the incidence of POD. Additionally, the prevalence of POD in our study was relatively low compared to previous retrospective studies. Previous retrospective studies reported an incidence rate of 9.3–18% in patients who underwent lumbar decompression or lumbar fusion surgery [19,24,25]. These differences were dependent on the diagnostic method used. In our study, the patients were diagnosed via psychiatric consultations when the symptoms developed. Some patients with milder symptoms maight have been missed during this process.

Previous reports investigating the risk factors for POD after spinal surgery have been published. The number of medications, medications with anticholinergic activity, and anti-psychotic medications used are related to POD [26,27,28]. Among the drugs used preoperatively, the use of anti-psychotics, corticosteroids, and antihistamines was significantly different between the POD and non-POD groups. We identified the risk factors of POD through a multivariate logistic regression analysis, and only anti-psychotic drugs, among the preoperatively used medications, were significantly associated with POD. The relationship between the use of steroids and antihistamines in the development of POD is unclear [28]. Our results were not different from those of previous studies. Additionally, a preoperative cognitive impairment is associated with POD. Previous meta-analyses showed that Mini-Mental State Examination scores are significantly lower in the POD group than those in the non-POD group [22]. However, we lacked information on preoperative cognitive functions that might increase the risk of POD, which was a limitation of this study.

Delirium is a neurobehavioral syndrome caused by the dysregulation of neuronal activity secondary to systemic disturbances. Several mechanisms have been proposed, involving neuroinflammation, neuronal aging, oxidative stress, neurotransmitter deficiency, neuroendocrine disruption, diurnal dysregulation, and network dysconnectivity [6]. Recent animal studies strongly suggest that POD may be caused by neuroinflammation in the central nervous system resulting from surgery-induced systematic inflammation [29,30]. Other data indicate that neuroinflammation is caused by excess levels of inflammatory cytokines (secreted by activated microglia) when the homeostasis of the central nervous system is disturbed, which may contribute to POD [31]. Some clinical reports showed that POD is associated with an increased immune-inflammatory biomarkers, including ILs and other cytokines [7,32,33]. A review of the basic and clinical studies concluded that immune-inflammatory biomarkers increase in subjects with delirium, suggesting that neuro-immune pathways play a role in the disorder [33]. In addition, evidence from meta-analyses suggests that POD is correlated with the concentrations of peripheral and CSF inflammatory markers, CRP, IL-6, -8, -10, and -1 beta; S-100 calcium-binding protein beta subunit; neuron-specific enolase; and TNF-α [9,10]. Some of these markers, such as CRP and IL-6, also play roles in POD [10], while others, such as IL-6 and neopterin (when present at high levels), have been associated with POD [34]. However, inflammatory cytokines are not routinely measured, as the means to do so are not widely available, expensive, and time-consuming, although increased inflammatory cytokine levels are associated with the development of POD.

Peripheral blood cells can reflect the inflammatory status of a patient. A series of hematological inflammatory biomarkers, including the NLR, MLR, and PLR, have been investigated in various diseases. Previous studies reported that the NLR increases in elderly patients with delirium [12], while an increase in the PLR was associated with the development of delirium in critically ill patients [35]. Several retrospective observational studies have shown that an increased NLR is associated with the development of POD after total hip arthroplasty or head and neck free-flap reconstruction [16,17]. Another study demonstrated that a preoperatively higher NLR, mean platelet volume, and platelet distribution width were each associated with a higher risk of POD in patients undergoing esophagectomy after adjusting for possible confounding factors [15]. The NLR results of our study were similar to those of previous ones.

In addition to the NLR, the MLR was higher in our patients with POD than without POD. The MLR reflects the balance between innate (monocyte) and adaptive (lymphocyte) immunity and serves as a simple indicator of immune status and inflammation. Thus, the MLR has been suggested to be able to predict the prognosis in various diseases, including cardiovascular disease, cancers, and neurological disorders [36,37,38,39,40,41,42,43]. However, to the best of our knowledge, no study has reported the predictive value of the MLR for POD.

Transient elevations in the serum white blood cell count and decreases in the platelet count are normal physiological responses to inflammation. Thus, the NLR, PLR, and the platelet-to-WBC ratio (PWR) are readily available inflammatory markers. Lower preoperative PLR and PWR levels are associated with POD after cardiac surgery [18]. However, no difference in the preoperative PLR was detected between the POD and non-POD groups in our study. More research is needed to understand the association between the PLR and POD.

In addition to hematological inflammatory markers, CRP and albumin (positive and negative acute phase reactants, respectively) are commonly used to measure inflammatory activity. The CAR has recently been considered as a more useful indicator of sepsis than CRP or albumin alone [44]. Additionally, the CAR is more sensitive and specific for predicting the systemic inflammatory state and prognosis than either CRP or albumin alone, for various clinical conditions [42,43,44,45,46,47,48]. A prospective study enrolled POD patients who were assessed daily within 7 postoperative days. The preoperative CAR level (OR: 3.04) was an independent risk factor for POD in elderly subjects undergoing total joint arthroplasty [49]. In another retrospective study, patients undergoing transcatheter aortic valve replacement were analyzed and the CAR was a promising inflammatory parameter to predict POD [50]. As few studies have been performed on the association between the CAR and POD, more research is needed before clinical applications.

Preoperative inflammatory markers, including the NLR, MLR, and CAR, as rapidly measurable and inexpensive biomarkers for systemic inflammation, may be useful for predicting POD, which is a multifactorial condition; some inflammatory biomarkers are common among the proposed pathomechanisms. In addition, the association between the preoperative blood levels of inflammatory mediators and POD may be affected by the type of surgery or specific mediators [9,10]. Thus, prospective studies including various types of surgeries are required to demonstrate the usefulness of the NLR, MLR, and CAR for POD in clinical practice.

This study had several limitations. First, it used a retrospective design, and the data were from subjects who visited a single university medical center with five hospitals. Therefore, generalizing the results is difficult and the possibility of a selection bias must be considered. We analyzed 11 years of data from the Smart CDW. The long recruitment period allowed for changes in surgical techniques and anesthesia management, and their impacts cannot be overlooked. In addition, clinical information regarding the patients’ preoperative cognitive function, as assessed using the Mini-Mental State Examination among other neuropsychology tests, was not collected. We also lacked information regarding the use of NSAIDs, which could affect the inflammatory markers. The relatively low incidence of POD in our study might have affected the results, so they should be interpreted with caution. Additionally, POD was assessed daily using the short form of the Korean Nursing Delirium Screening Scale, by the nurse involved in this study. Patients suspected to have developed POD as assessed by the nurse received psychiatric counseling and the POD diagnosis was confirmed. Thus, patients who had hypoactive delirium or mild symptoms of POD, and did not consult a psychiatrist, might have been excluded, which could explain the lower incidence of POD in our study. Additionally, we collected the medical records, including nursing notes and request notes for consultation with a psychiatrist, within 48 h after surgery. When the psychiatrist responded to the consultation request, POD was confirmed, and the patient was included in the POD group. It was difficult to evaluate the overall POD due to the short evaluation period. Additionally, we could not explore the associations between the inflammatory markers and the clinical characteristics after POD developed, as the laboratory inflammatory marker data were collected only once during the preoperative period. Nevertheless, this was the first study to evaluate the association between easily measured preoperative inflammatory markers, including the NLR, MLR, CAR, and POD, in patients undergoing lumbar spinal fusion surgery. Although the results suggest neuroinflammation as a mechanism of POD, more detailed studies are needed for confirmation.

## 5. Conclusions

In conclusion, this retrospective study using data from the CDW obtained over an 11-year period demonstrated that elevated preoperative inflammatory markers, including the NLR, MLR, and CAR, were associated with the development of POD. Thus, a preoperative inflammatory state is a potential pathophysiological mechanism of POD. Additional prospective studies are needed to further investigate inflammatory markers of POD.

## Figures and Tables

**Figure 1 jcm-11-04085-f001:**
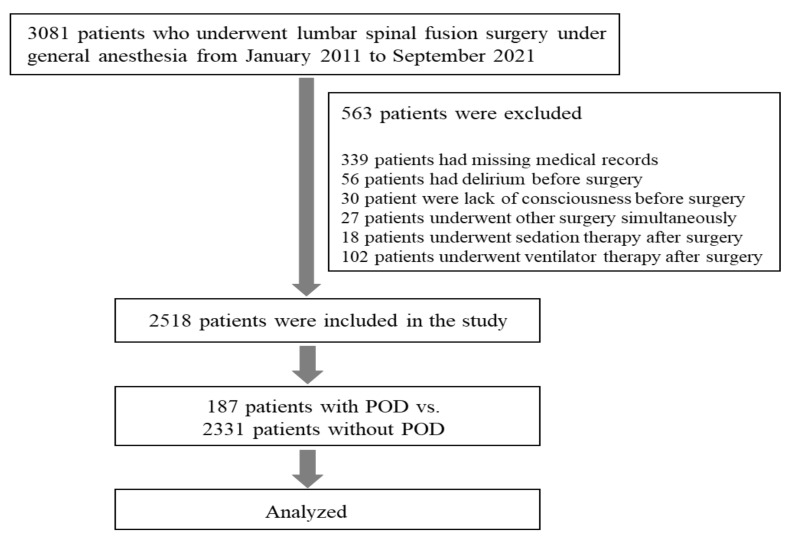
Flow chart of the enrollment process. POD, postoperative delirium.

**Figure 2 jcm-11-04085-f002:**
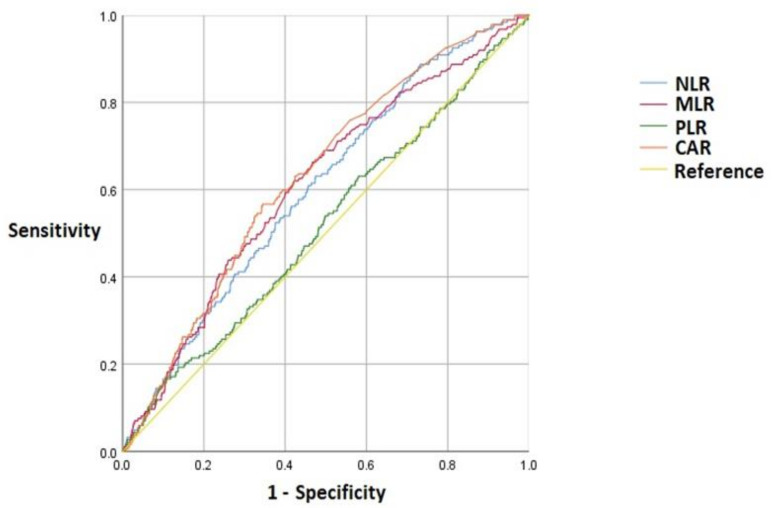
ROC curve showing the discriminative ability of the inflammatory markers (NLR, MLR, PLR, and CAR) in the POD and non-POD groups. ROC, receiver operating characteristic; NLR, neutrophil-to-lymphocyte ratio; MLR, monocyte-to-lymphocyte ratio; PLR, platelet-to-lymphocyte ratio; CAR, CRP-to-albumin ratio.

**Table 1 jcm-11-04085-t001:** Preoperative, intraoperative, and postoperative risk factors of the non-POD and POD groups.

Variable	non-POD (*n* = 2331)	POD (*n* = 187)	*p* Value
Age, *n* (median, IQR)	65 (56, 72)	74 (67, 78)	<0.001
Male, *n* (%)	1011 (43.4)	96 (51.3)	0.035
Obesity (BMI > 29.9), *n* (%)	195 (8.4)	9 (4.8)	0.087
ASA physical status > 2, *n* (%)	715 (30.7)	112 (59.9)	<0.001
Emergency surgery, *n* (%)	126 (5.4)	7 (3.7)	0.328
HTN, *n* (%)	1197 (51.4)	106 (56.7)	0.16
DM, *n* (%)	571 (24.5)	56 (29.9)	0.097
Heart disease, *n* (%)	267 (11.5)	35 (18.7)	0.003
Stroke, *n* (%)	135 (5.8)	20 (10.7)	0.007
Cancer, *n* (%)	182 (7.8)	13 (7.0)	0.674
Dyslipidemia, *n* (%)	430 (18.4)	21 (11.2)	0.013
Parkinson’s disease, *n* (%)	19 (0.8)	9 (4.8)	<0.001
Dementia, *n* (%)	21 (0.9)	7 (3.7)	<0.001
Depression, *n* (%)	60 (2.6)	8 (4.3)	0.167
Kidney disease, *n* (%)	106 (4.5)	11 (5.9)	0.404
Liver disease, *n* (%)	79 (3.4)	5 (2.7)	0.6
Insomnia, *n* (%)	144 (6.2)	17 (9.1)	0.117
Sleep disorder, *n* (%)	137 (5.9)	16 (8.6)	0.14
Alcohol, *n* (%)	585 (25.1)	32 (17.1)	0.015
Smoking, *n* (%)	384 (16.5)	32 (17.1)	0.821
Preoperative used drugs			
Calcium channel blockers, *n* (%)	1014 (43.5)	81 (43.3)	0.961
Diuretics, *n* (%)	230 (9.9)	22 (11.8)	0.405
Beta blockers, *n* (%)	192 (8.2)	13 (7.0)	0.536
ACE inhibitors, *n* (%)	10 (0.4)	1 (0.5)	0.833
Angiotensin receptor blockers, *n* (%)	139 (6.0)	11 (5.9)	0.964
Other antihypertensives, *n* (%)	29 (1.2)	1 (0.5)	0.39
Miscellaneous CV drugs, *n* (%)	127 (5.4)	13 (7.0)	0.388
Anti-depressants, *n* (%)	77 (3.3)	4 (2.1)	0.385
Hypnotics, *n* (%)	718 (30.8)	67 (35.8)	0.153
Anti-psychotics, *n* (%)	168 (7.2)	33 (17.6)	<0.001
Opioids, *n* (%)	2317 (99.4)	186 (99.5)	0.91
Corticosteroids, *n* (%)	648 (27.8)	68 (36.4)	0.012
Antihistamines, *n* (%)	815 (35.0)	95 (50.8)	<0.001
H_2_ blockers, *n* (%)	689 (29.6)	54 (28.9)	0.844
Intra and postoperative factors			
Postop. ICU care, *n* (%)	339 (14.5)	90 (48.1)	<0.001
Patient-controlled analgesia, *n* (%)	2236 (95.9)	180 (96.3)	0.825
Operation time, hours (median, IQR)	3.6 (2.8, 4.6)	3.9 (2.9, 5.1)	0.004
Surgical range, level (median, IQR)	2 (1, 2)	2 (1, 3)	0.002
Fluid balance (input–output), mL (median, IQR)	1.0 (0.6, 1.6)	1.1 (0.6, 1.8)	0.074

POD, postoperative delirium; IQR, interquartile range; BMI, body mass index; ASA, American Society of Anesthesiology physical status; ICU, intensive-care unit; ACE, angiotensin converting enzyme; CV, cerebrovascular.

**Table 2 jcm-11-04085-t002:** Differences in preoperative inflammatory markers between the non-POD and POD groups.

	non-POD (*n* = 2331)	POD (*n* = 187)	*p* Value
NLR, median (IQR)	2.17 (1.54, 3.29)	2.68 (1.90, 4.19)	<0.001
MLR, median (IQR)	0.23 (0.17, 0.33)	0.29 (0.21, 0.39)	<0.001
PLR, median (IQR)	137.88 (105.04, 182.98)	140.98 (103.86, 185.35)	0.465
CAR, median (IQR)	0.26 (0.22, 0.67)	0.47 (0.25, 1.89)	<0.001

POD, postoperative delirium; NLR, neutrophil-to-lymphocyte ratio; MLR, monocyte-to-lymphocyte ratio; PLR, platelet-to-lymphocyte ratio; CAR, CRP-to-albumin ratio.

**Table 3 jcm-11-04085-t003:** Unadjusted odds ratios for the effect of the inflammatory markers on developing POD.

	IQR of Inflammatory Markers			
	NLR Q1	NLR Q2	NLR Q3	NLR Q4
uOR	reference	2.38	2.73	3.28
95% CI		1.41–4.0	1.63–4.58	1.98–5.45
*p* value		0.001	<0.001	<0.001
	MLR Q1	MLR Q2	MLR Q3	MLR Q4
uOR	reference	1.21	1.88	3.0
95% CI		0.72–2.02	1.17–3.04	1.92–4.7
*p* value		0.472	0.009	<0.001
	PLR Q1	PLR Q2	PLR Q3	PLR Q4
uOR	reference	0.8	1.09	1.0
95% CI		0.52–1.24	0.72–1.64	0.66–1.52
*p* value		0.314	0.683	>0.999
	CAR Q1	CAR Q2	CAR Q3	CAR Q4
uOR	reference	1.78	2.51	3.38
95% CI		1.06–2.97	1.56–4.04	2.14–5.33
*p* value		0.029	<0.001	<0.001

IQR, interquartile range; uOR, unadjusted odds ratio; CI, confidence interval; NLR, neutrophil-to-lymphocyte ratio; MLR, monocyte-to-lymphocyte ratio; PLR, platelet-to-lymphocyte ratio; CAR, CRP-to-albumin ratio.

**Table 4 jcm-11-04085-t004:** Multivariate logistic regression analysis showing the effect of inflammatory markers and other variables on the likelihood of POD development.

Variable	aOR (95% CI)	*p* Value
IQR of Inflammatory Markers		
IQR NLR Q2	2.26 (1.24–4.14)	0.008
IQR NLR Q3	2.47 (1.3–4.72)	0.006
IQR NLR Q4	2.86 (1.38–5.92)	0.005
IQR MLR Q2	0.87 (0.48–1.56)	0.636
IQR MLR Q3	1.07 (0.6–1.91)	0.819
IQR MLR Q4	1.46 (0.77–2.75)	0.241
IQR PLR Q2	0.5 (0.3–0.84)	0.008
IQR PLR Q3	0.7 (0.42–1.16)	0.168
IQR PLR Q4	0.48 (0.27–0.85)	0.011
IQR CAR Q2	0.92 (0.52–1.63)	0.782
IQR CAR Q3	1.53 (0.9–2.59)	0.116
IQR CAR Q4	1.6 (0.94–2.72)	0.085
Other variables		
Age	1.05 (1.03–1.07)	<0.001
Male	1.8 (1.22–2.65)	0.003
Obesity (BMI > 29.9)	0.72 (0.33–1.53)	0.39
ASA physical status > 2	2.07 (1.37–3.13)	<0.001
Emergency surgery	0.92 (0.38–2.23)	0.852
HTN	0.77 (0.53–1.12)	0.174
DM	0.95 (0.64–1.4)	0.779
Heart disease	0.71 (0.44–1.16)	0.171
Stroke	1.04 (0.58–1.88)	0.885
Cancer	0.52 (0.28–0.99)	0.047
Dyslipidemia	0.66 (0.4–1.11)	0.12
Parkinson’s disease	4.46 (1.76–11.29)	0.002
Dementia	3.8 (1.4–10.31)	0.009
Depression	2.64 (1.14–6.16)	0.024
Kidney disease	0.78 (0.37–1.61)	0.496
Liver disease	0.48 (0.17–1.4)	0.179
Insomnia	0.9 (0.07–12.27)	0.937
Sleep disorder	1.64 (0.11–23.51)	0.717
Alcohol	0.66 (0.4–1.08)	0.097
Smoking	1.1 (0.66–1.85)	0.714
Calcium channel blockers	0.85 (0.6–1.21)	0.37
Diuretics	0.76 (0.43–1.32)	0.328
Beta blockers	0.79 (0.4–1.55)	0.486
ACE inhibitors	0.95 (0.08–11.84)	0.97
Angiotensin receptor blockers	1.13 (0.54–2.38)	0.741
Other antihypertensives	0.25 (0.03–2.31)	0.222
Miscellaneous CV drugs	1.51 (0.75–3.03)	0.247
Antidepressants	0.64 (0.21–1.93)	0.424
Hypnotic sedatives	1.07 (0.71–1.61)	0.749
Antipsychotics	3.52 (2.08–5.97)	<0.001
Opioids	0.69 (0.08–6.09)	0.734
Corticosteroids	1.16 (0.8–1.68)	0.43
Antihistamine antiallergics	1.41 (0.95–2.11)	0.089
H2 receptor antagonist	1.08 (0.71–1.62)	0.729
Operation time	1.11 (0.97–1.26)	0.118
Surgical range	1.13 (0.96–1.32)	0.146
Fluid balance (input–output)	0.91 (0.75–1.12)	0.383
Postop. ICU care	3.83 (2.59–5.65)	<0.001
Patient-controlled analgesia	1.13 (0.45–2.81)	0.796

IQR, interquartile range; aOR, adjusted odds ratio; CI, confidence interval; NLR, neutrophil-to-lymphocyte ratio; MLR, monocyte-to-lymphocyte ratio; PLR, platelet-to-lymphocyte ratio; CAR, CRP-to-albumin ratio.

**Table 5 jcm-11-04085-t005:** Optimal cutoff values of the preoperative inflammatory markers for discriminating between the POD and non-POD groups.

Inflammatory Markers	Cutoff Value	AUC (95% CI)	Sensitivity	Specificity	*p* Value
NLR	2.26	0.60 (0.56–0.64)	0.63	0.53	<0.001
MLR	0.26	0.61 (0.57–0.65)	0.62	0.58	<0.001
PLR	233.90	0.52 (0.47–0.56)	0.16	0.90	0.464
CAR	0.38	0.63 (0.59–0.67)	0.57	0.66	<0.001

AUC, area under the curve; NLR. neutrophil-to-lymphocyte ratio; MLR, monocyte-to-lymphocyte ratio; PLR, platelet-to-lymphocyte ratio; CAR, CRP-to-albumin ratio.

## Data Availability

Not applicable.

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
