# Peer review of "Preoperative Inflammatory Markers and the Risk of Postoperative Delirium in Patients Undergoing Lumbar Spinal Fusion Surgery"

_jcm, 2022, doi:10.3390/jcm11144085_

Round 1
Reviewer 1 Report
Thanks for the revision, I don't have any further comments.
Author Response
July 13, 2022
Reviewer 1
Journal of Clinical Medicine
Dear Reviewer 1,
Please find attached a revised version of our manuscript, “Preoperative Inflammatory Markers and the Risk of Postoperative Delirium in Patients Undergoing Lumbar Spinal Fusion Surgery” (jcm- 1811184).
We thank you for your suggestions regarding the original version of our paper; most of the suggested changes have been incorporated into the revision.
All of the revisions are described in detail in the order mentioned in the review, following the reviewer’s comments in italics. We believe that the revisions have greatly improved the manuscript and hereby submit the revised version for consideration for publication.
Comments to author:
Thanks for the revision, I don't have any further comments.
We thank the reviewer for the comments and specific suggestions, which have helped us to improve our manuscript.
We are grateful for the constructive comments provided during the review process. We believe that our paper has been improved by these suggestions.
Yours faithfully,
Jong-Hee Sohn, M.D. Ph.D.
Department of Neurology, Chuncheon Sacred Heart Hospital, Hallym University College of Medicine, 77 Sakju-ro, Chuncheon-si, Gangwon-do, 24253, Republic of Korea
Tel: +82-33-252-9970, Fax: +82-33-241-8063
E-mail: deepfoci@hallym.or.kr

Reviewer 2 Report
The authors determined the associations between preoperative blood inflammatory markers and POD in patients undergoing lumbar spinal fusion surgery. This is a retrospective study and of the 3,081 enrolled patients, 187 developed POD. A significant increase in the NLR, MLR and CAR was observed in the POD group. Multivariate analysis showed that the second, third and highest quartiles of the NLR were significantly associated with the development of POD. The ROC curve revealed that the discriminative ability of the NLR, MLR, and CAR for predicting POD development was low, but marginally acceptable. Some minor concerns were listed below:
1 The prevalence of POD in the retrospective study was relatively low compared to previous studies. The authors explained that a retrospective assessment of POD may have led to underestimation of the incidence of POD. I suggest that the authors shoud compared the prevalence of POD with the previous retrospective studies.
2 In this study, clinical information regarding the patients’ preoperative cognitive function, as assessed using the Mini-Mental State Examination among other neuropsychology tests, was not collected. The authors should describe and discuss how the testsof preoperative cognitive function will affect the incidence of POD in the previous studies.
3 In the two groups, the use of corticosteroids was significantly different. The authors shoud discuss the role of corticosteriods in the development of POD.
4 In patients with lumbar fusion surgery, most regularly received the oral pills of NSAIDs for the pain treatment. However, the author did not list the difference of use of NSAIDs in patients. The use of NSAIDs may affect the inflammatory levels.
Author Response
July 13, 2022
Reviewer 2
Journal of Clinical Medicine
Dear Reviewer 2,
Please find attached a revised version of our manuscript, “Preoperative Inflammatory Markers and the Risk of Postoperative Delirium in Patients Undergoing Lumbar Spinal Fusion Surgery” (jcm- 1811184).
We thank you for your suggestions regarding the original version of our paper; most of the suggested changes have been incorporated into the revision.
All of the revisions are described in detail in the order mentioned in the review, following the reviewer’s comments in italics. We believe that the revisions have greatly improved the manuscript and hereby submit the revised version for consideration for publication.
Comments to author:
The authors determined the associations between preoperative blood inflammatory markers and POD in patients undergoing lumbar spinal fusion surgery. This is a retrospective study and of the 3,081 enrolled patients, 187 developed POD. A significant increase in the NLR, MLR and CAR was observed in the POD group. Multivariate analysis showed that the second, third and highest quartiles of the NLR were significantly associated with the development of POD. The ROC curve revealed that the discriminative ability of the NLR, MLR, and CAR for predicting POD development was low, but marginally acceptable. Some minor concerns were listed below:
We thank the reviewer for these comments and suggestions, which have helped us to improve our manuscript.
- The prevalence of POD in the retrospective study was relatively low compared to previous studies. The authors explained that a retrospective assessment of POD may have led to underestimation of the incidence of POD. I suggest that the authors should compared the prevalence of POD with the previous retrospective studies.
The incidence of POD varied depending on the type of surgery. The prevalence of POD after spinal surgery ranged from 3.8 to 40.4% and from 0.84 to 27.6% in two previous meta-analyses. The prevalence of POD after lumbar spinal fusion surgery was 7.4% in our study. This retrospective assessment of POD may have led to underestimate of its incidence. Additionally, the prevalence of POD in our study was relatively low compared to previous retrospective studies. Previous retrospective studies reported an incidence rate of 9.3–18% in patients who underwent lumbar decompression or lumbar fusion surgery. These differences in the incidence of POD were dependent on the diagnostic method used. In our study, patients were diagnosed via psychiatric consultations when symptoms developed, to provide a solid diagnosis. Some patients with milder symptoms may have been missed during this process.
We have revised the Discussion, as follows:
Additionally, the prevalence of POD in our study was relatively low compared to previous retrospective studies. Previous retrospective studies reported an incidence rate of 9.3 – 18% in patients who underwent lumbar decompression or lumbar fusion surgery [24-26]. These differences were dependent on the diagnostic method used. In our study, patients were diagnosed via psychiatric consultations when symptoms developed. Some patients with milder symptoms may have been missed during this process.
(page 9, lines 231– page 9, lines 237)
We have also added these citations.
- Jiang, X.; Chen, D.; Lou, Y.; Li, Z. Risk factors for postoperative delirium after spine surgery in middle- and old-aged patients. Aging Clin Exp Res. 2017, 29(5):1039-44. doi: 10.1007/s40520-016-0640-4.
- Susano, M.J.; Scheetz, S.D.; Grasfield, R.H.; Cheung,D.; Xu, X.; Kang, J.D.; Smith, T.R.; Lu, Y.; Groff, M.W.; Chi, J.H.; Crosby, G.; Culley, D.J.. Retrospective analysis of perioperative variables associated with postoperative delirium and other adverse outcomes in older patients after spine surgery. J Neurosurg Anesthesiol. 2019 Oct;31(4):385-391. doi: 10.1097/ANA.0000000000000566.
- Fineberg, S.J.; Nandyala, S.V.; Marquez-Lara, A.; Oglesby, M.; Patel, A.A.; Singh, K. Incidence and risk factors for postoperative delirium after lumbar spine surgery (Phila Pa 1976). Spine 2013, 38, 1790–-6. doi: 10.1097/BRS.0b013e3182a0d507
- In this study, clinical information regarding the patients’ preoperative cognitive function, as assessed using the Mini-Mental State Examination among other neuropsychology tests, was not collected. The authors should describe and discuss how the tests of preoperative cognitive function will affect the incidence of POD in the previous studies.
We agree. Preoperative cognitive impairment is associated with POD. The results of previous studies showed that the Mini-Mental State Examination score is significantly associated with POD. However, this study was retrospective and lacked information on preoperative cognitive function, such as the Mini-Mental State Examination or other neuropsychology tests that might indicate an increased risk of POD; this was a limitation of the study.
We have revised the Discussion as follows.
Additionally, preoperative cognitive impairment is associated with POD. Previous me-ta-analyses showed that Mini-Mental State Examination scores are significantly lower in the POD group than those in the non-POD group [22]. However, we lacked information on preoperative cognitive function, that might increase the risk of POD, which was a limitation of this study
(page 9, lines 246 – page 9, lines 250)
- In the two groups, the use of corticosteroids was significantly different. The authors should discuss the role of corticosteriods in the development of POD.
Previous studies have reported that polypharmacy, medications with anticholinergic activity, and anti-psychotic medications are related to POD. The involvement of steroids and antihistamines in the development of POD is less clear. The preoperative use of anti-psychotics, corticosteroids, and antihistamines was significantly different between the POD and non-POD groups in our study. We identified the risk factors of POD in patients undergoing spinal fusion surgery through multivariate logistic regression analysis, and only anti-psychotics, among the medications used preoperatively, were significantly associated with POD (OR 3.52, 95% CI 2.08–5.97; p < 0.001). Our results were not different from those of previous studies.
Thus, we have revised the Discussion as follows.
Previous reports investigating the risk factors for POD after spinal surgery have been published. The number of medications, medications with anticholinergic activity, and an-ti-psychotic medications are related to POD [27-29]. Among the drug used preoperatively, the use of anti-psychotics, corticosteroids, and antihistamines was significantly different between the POD and non-POD groups. We identified the risk factors of POD through multivariate logistic regression analysis, and only anti-psychotic drug among the pre-operatively used medications, were significantly associated with POD. The relationship between the use of steroids, and antihistamines in the development of POD is unclear [29]. Our results were not different from those of previous studies.
(page 9, lines 238 – page 9, lines 246)
We have added these citations.
- Kassie, G.M.; Nguyen, T.A.; Kalisch Ellett, L.M.; Pratt, N.L.; Roughead, E.E. Preoperative medication use and postoperative delirium: A systematic review. BMC Geriatr. 2017, 17, 298. [CrossRef]
- Bilotta, F.; Lauretta, M.P.; Borozdina, A.; Mizikov, V.M.; Rosa, G. Postoperative delirium: Risk factors, diagnosis and perioperative care. Minerva Anestesiol. 2013, 79, 1066–1076.
- Clegg, A.; Young, J.B. Which medications to avoid in people at risk of delirium: A systematic review. Age Ageing 2011, 40, 23–29.
- In patients with lumbar fusion surgery, most regularly received the oral pills of NSAIDs for the pain treatment. However, the author did not list the difference of use of NSAIDs in patients. The use of NSAIDs may affect the inflammatory levels.
We agree. Unfortunately, we did not include NSAID use data in our basic data. It is impossible to obtain data within a short time from an institution that manages a clinical data warehouse in a limited time. NSAID data cannot be included given the time limit of the revision. Thus, we describe this as a limitation, as follows:
We also lacked information regarding the use of NSAIDs, which could affect the inflammatory markers; this was a limitation of the study.
(page 11, lines 323 – page 11, lines 324)
We have addressed all of the issues raised by the reviewers. We are grateful for the constructive comments provided during the review process. We believe that our paper has been improved by these suggestions.
Yours faithfully,
Jong-Hee Sohn, M.D. Ph.D.
Department of Neurology, Chuncheon Sacred Heart Hospital, Hallym University College of Medicine, 77 Sakju-ro, Chuncheon-si, Gangwon-do, 24253, Republic of Korea
Tel: +82-33-252-9970, Fax: +82-33-241-8063
E-mail: deepfoci@hallym.or.kr

This manuscript is a resubmission of an earlier submission. The following is a list of the peer review reports and author responses from that submission.
Round 1
Reviewer 1 Report
This is a nicely written manuscript on factors influencing POD, based on a retrospective analysis on existing data.
I like the manuscript, however some comments / suggestions / questions remain:
o Abstract: I think the AUC measures more the ability of a variable to discriminate
o Introduction: It was unclear to me, why the authors focus on spinal surgeries. Is there an hpyothesis, that the effect of inflammatory markers depends on the surgery type?
o Why was age used in a binary fashion (>= 70 years, see Section 2.2.). Isn't age one of the main predictors for POD? I think it could make sense to treat this variable as continuous
o Statistical analysis: If I understood correctly, matching was performed to select the controls. In this case, it should be necessary to apply paired test because of the incorporated dependency between cases and controls.
o Statistical analysis: How was the propensity score constructed, which variables entered?
o Statistical analysis: How exactly were optimal cutpoints determined? Of course we have a ROC displaying all the potential cutpoints, but there are different approaches on which is then the optimal one.
Author Response
June 8, 2022
Reviewer 1
Journal of Clinical Medicine
Dear Reviewer 1,
Please find attached a revised version of our manuscript, “Preoperative Inflammatory Markers and the Risk of Postoperative Delirium in Patients Undergoing Lumbar Spinal Fusion Surgery” (jcm- 1753872).
We thank you for your thoughtful suggestions regarding the original version of our paper; most of the suggested changes have been incorporated into the revision.
All revisions are described in detail in the order mentioned in the review, following the reviewer’s critique in italics. We believe that the revisions have greatly improved the manuscript and hereby submit the revised version for your consideration for publication.
Comments to author:
This is a nicely written manuscript on factors influencing POD, based on a retrospective analysis on existing data. I like the manuscript, however some comments / suggestions / questions remain:
We thank the reviewer for these comments and specific suggestions, which have improved our manuscript.
o Abstract: I think the AUC measures more the ability of a variable to discriminate
We have revised the sentence in the Abstract, as follows:
Receiver operating characteristic curve analysis showed that the discriminative ability of the NLR, MLR, and CAR for predicting POD was close to good.
(page 1, lines 27 – page 1, lines 29)
o Introduction: It was unclear to me, why the authors focus on spinal surgeries. Is there an hypothesis, that the effect of inflammatory markers depends on the surgery type?
In the meta-analysis, standardized mean differences of preoperative inflammatory markers, such as CRP and IL-6 levels, were significantly positive correlated with surgery type. Another meta-analysis reported that the association between preoperative blood levels of inflammatory mediators and postoperative delirium (POD) may be influenced by the type of surgery. It is likely that there is a potential contribution of special circumstances associated with certain types of surgery to the etiology of POD. We have added this point to the Introduction as follows:
The standardized mean difference of preoperative inflammatory markers, such as CRP and IL-6 levels, is significantly positive correlated with surgery type [10]. Another study reported that the association between preoperative blood levels of inflammatory mediators and POD may be affected by the type of surgery. The special circumstances associated with certain types of surgery may contribute to the etiology of POD.
(page 2, lines 67 – page 2, lines 72)
o Why was age used in a binary fashion (>= 70 years, see Section 2.2.). Isn't age one of the main predictors for POD? I think it could make sense to treat this variable as continuous
We performed the statistical analysis with age as a continuous variable. We have revised Table 1, as follows:
Table 1. Preoperative, intraoperative, and postoperative risk factors of the non-POD and POD groups before PSM and after PSM.
Variable |
Before PSM |
After PSM |
||||||||||||
non-POD (n=2,331) |
POD (n=187) |
ASD |
non-POD (n=374) |
POD (n=187) |
ASD |
|||||||||
Age, n (median, IQR) |
65 (56, 72) |
74 (67, 78) |
0.77 |
73 (67, 78) |
74 (67, 78) |
0.05 |
||||||||
Male, n (%) |
1007 (43.4) |
96 (51.3) |
0.16 |
178 (47.6) |
96 (51.3) |
0.07 |
||||||||
Obesity (BMI>29.9), n (%) |
195 (8.4) |
9 (4.8) |
0.17 |
12 (3.2) |
9 (4.8) |
0.07 |
||||||||
ASA physical status>2, n (%) |
714 (30.7) |
112 (59.9) |
0.59 |
235 (62.8) |
112 (59.9) |
0.06 |
||||||||
Emergency surgery, n (%) |
126 (5.4) |
7 (3.7) |
0.09 |
14 (3.7) |
7 (3.7) |
0.00 |
||||||||
HTN, n (%) |
1194 (51.4) |
106 (56.7) |
0.11 |
213 (57.0) |
106 (56.7) |
0.01 |
||||||||
DM, n (%) |
569 (24.5) |
56 (29.9) |
0.12 |
112 (29.9) |
56 (29.9) |
0.00 |
||||||||
Heart disease, n (%) |
267 (11.5) |
35 (18.7) |
0.18 |
64 (17.1) |
35 (18.7) |
0.04 |
||||||||
Stroke, n (%) |
135 (5.8) |
20 (10.7) |
0.16 |
40 (10.7) |
20 (10.7) |
0.00 |
||||||||
Cancer, n (%) |
181 (7.8) |
13 (7.0) |
0.03 |
29 (7.8) |
13 (7.0) |
0.03 |
||||||||
Dyslipidemia, n (%) |
428 (18.4) |
21 (11.2) |
0.23 |
47 (12.6) |
21 (11.2) |
0.04 |
||||||||
Parkinson’s disease, n (%) |
19 (0.8) |
9 (4.8) |
0.19 |
15 (4.0) |
9 (4.8) |
0.04 |
||||||||
Dementia, n (%) |
21 (0.9) |
7 (3.7) |
0.15 |
14 (3.7) |
7 (3.7) |
0.00 |
||||||||
Depression, n (%) |
60 (2.6) |
8 (4.3) |
0.08 |
15 (4.0) |
8 (4.3) |
0.01 |
||||||||
Kidney disease, n (%) |
106 (4.6) |
11 (5.9) |
0.06 |
23 (6.1) |
11 (5.9) |
0.01 |
||||||||
Liver disease, n (%) |
79 (3.4) |
5 (2.7) |
0.05 |
10 (2.7) |
5 (2.7) |
0.00 |
||||||||
Insomnia, n (%) |
143 (6.2) |
17 (9.1) |
0.10 |
36 (9.6) |
17 (9.1) |
0.02 |
||||||||
Sleep disorder, n (%) |
136 (5.9) |
16 (8.6) |
0.10 |
35 (9.4) |
16 (8.6) |
0.03 |
||||||||
Alcohol, n (%) |
583 (25.1) |
32 (17.1) |
0.21 |
57 (15.2) |
32 (17.1) |
0.05 |
||||||||
Smoking, n (%) |
383 (16.5) |
32 (17.1) |
0.02 |
55 (14.7) |
32 (17.1) |
0.06 |
||||||||
Preoperative used drugs |
||||||||||||||
Calcium channel blockers, n (%) |
1011 (43.5) |
81 (43.3) |
0.00 |
156 (41.7) |
81 (43.3) |
0.03 |
||||||||
Diuretics, n (%) |
230 (9.9) |
22 (11.8) |
0.06 |
37 (9.9) |
22 (11.8) |
0.06 |
||||||||
Beta blockers, n (%) |
190 (8.2) |
13 (7.0) |
0.05 |
22 (5.9) |
13 (7.0) |
0.04 |
||||||||
ACE inhibitors, n (%) |
10 (0.4) |
1 (0.5) |
0.01 |
3 (0.8) |
1 (0.5) |
0.04 |
||||||||
Angiotensin receptor blockers, n (%) |
139 (6.0) |
11 (5.9) |
0.00 |
23 (6.1) |
11 (5.9) |
0.01 |
||||||||
Other antihypertensives, n (%) |
29 (1.2) |
1 (0.5) |
0.10 |
1 (0.3) |
1 (0.5) |
0.04 |
||||||||
Miscellaneous CV drugs, n (%) |
126 (5.4) |
13 (7.0) |
0.06 |
25 (6.7) |
13 (7.0) |
0.01 |
||||||||
Anti-depressants, n (%) |
77 (3.3) |
4 (2.1) |
0.08 |
8 (2.1) |
4 (2.1) |
0.00 |
||||||||
Hypnotics, n (%) |
718 (30.9) |
67 (35.8) |
0.10 |
128 (34.2) |
67 (35.8) |
0.03 |
||||||||
Anti-psychotics, n (%) |
168 (7.2) |
33 (17.6) |
0.27 |
55 (14.7) |
33 (17.6) |
0.08 |
||||||||
Opioids, n (%) |
2308 (99.4) |
186 (99.5) |
0.01 |
371 (99.2) |
186 (99.5) |
0.04 |
||||||||
Corticosteroids, n (%) |
644 (27.7) |
68 (36.4) |
0.18 |
145 (38.8) |
68 (36.4) |
0.05 |
||||||||
Antihistamines, n (%) |
815 (35.1) |
95 (50.8) |
0.31 |
183 (48.9) |
95 (50.8) |
0.04 |
||||||||
H2 blockers, n (%) |
687 (29.6) |
54 (28.9) |
0.02 |
111 (29.7) |
54 (28.9) |
0.02 |
||||||||
Intra and postoperative factors |
||||||||||||||
Postop. ICU care, n (%) |
339 (14.6) |
90 (48.1) |
0.67 |
78 (20.9) |
90 (48.1) |
0.54 |
||||||||
Patient-controlled analgesia, n (%) |
2,228 (96.0) |
180 (96.3) |
0.02 |
365 (97.6) |
180 (96.3) |
0.07 |
||||||||
Operation time, hour (median, IQR) |
3.6 (2.8, 4.6) |
3.9 (2.9, 5.1) |
0.26 |
3.7 (2.8, 4.7) |
3.9 (2.9, 5.1) |
0.23 |
||||||||
Surgical range, level (median, IQR) |
2 (1, 2) |
2 (1, 3) |
0.23 |
2 (1, 2) |
2 (1, 3) |
0.15 |
||||||||
Fluid balance (input–output), mL (median, IQR) |
1.0 (0.6, 1.6) |
1.1 (0.6, 1.8) |
0.15 |
1.1 (0.7, 1.8) |
1.1 (0.6, 1.8) |
0.06 |
||||||||
o Statistical analysis: If I understood correctly, matching was performed to select the controls. In this case, it should be necessary to apply paired test because of the incorporated dependency between cases and controls.
The reason we used propensity score matching was to see whether there were any differences in the NMR, MLR, PLR, and CAR of those who developed POD and those who did not develop POD when other variables before surgery were controlled. Only the preoperative conditions and administered drugs were used for matching, and other variables during and after surgery were included in the calculation of the odds ratios. We performed propensity score matching and statistical analysis with modifications. We have revised the text in the Methods as follows:
Only preoperative conditions and the administrated drugs were used for matching, and other variables during and after surgery were additionally included in the odds ratio calculation.
(page 4, lines 153 – page 4, lines 155)
Data before matching were used to calculate the odds ratios and to conduct the receiver operating characteristic (ROC) curve analysis.
(page 4, lines 160 – page 4, lines 161)
Also, we have revised Table 2 and 3, as follows:
Table 2. Differences in the preoperative inflammatory markers between the non-POD and POD groups before and after PSM.
|
Before PSM |
P value |
After PSM |
P value |
|||
non-POD (n=2,322) |
POD (n=187) |
non-POD (n=374) |
POD (n=187) |
||||
NLR median (IQR) |
2.17 (1.55, 3.29) |
2.68 (1.90, 4.19) |
<0.001 |
2.26 (1.72, 3.36) |
2.68 (1.90, 4.19) |
0.002 |
|
MLR median (IQR) |
0.23 (0.17, 0.33) |
0.29 (0.21, 0.39) |
<0.001 |
0.26 (0.19, 0.36) |
0.29 (0.21, 0.39) |
0.049 |
|
PLR median (IQR) |
137.86 (105.02, 182.93) |
140.98 (103.86, 185.35) |
0.464 |
136.94 (104.89, 184.95) |
140.98 (103.86, 185.35) |
0.713 |
|
CAR median (IQR) |
0.26 (0.22, 0.68) |
0.47 (0.25, 1.89) |
<0.001 |
0.27 (0.24, 0.98) |
0.47 (0.25, 1.89) |
0.001 |
|
Table 3. Logistic regression analysis showing the effect of inflammatory markers on developing POD.
|
IQR of inflammatory markers |
|||
|
NLR Q1 |
NLR Q2 |
NLR Q3 |
NLR Q4 |
aOR |
reference |
2.28 |
2.48 |
2.88 |
95% CI |
1.25-4.16 |
1.3-4.73 |
1.39-5.96 |
|
p-value |
0.008 |
0.006 |
0.005 |
|
|
MLR Q1 |
MLR Q2 |
MLR Q3 |
MLR Q4 |
aOR |
reference |
0.87 |
1.07 |
1.47 |
95% CI |
0.48-1.57 |
0.6-1.92 |
0.78-2.76 |
|
p-value |
0.651 |
0.809 |
0.236 |
|
|
PLR Q1 |
PLR Q2 |
PLR Q3 |
PLR Q4 |
aOR |
reference |
0.5 |
0.7 |
0.48 |
95% CI |
0.3-0.84 |
0.42-1.16 |
0.27-0.85 |
|
p-value |
0.008 |
0.169 |
0.011 |
|
|
CAR Q1 |
CAR Q2 |
CAR Q3 |
CAR Q4 |
aOR |
reference |
0.92 |
1.55 |
1.56 |
95% CI |
0.52-1.63 |
0.92-2.63 |
0.91-2.66 |
|
p-value |
0.775 |
0.101 |
0.103 |
o Statistical analysis: How was the propensity score constructed, which variables entered?
We used 1:2 propensity score matching without replacement. The estimation algorithm was logistic regression and the matching algorithm was the nearest neighbor. The covariates for matching included age, sex, obesity, ASA physical status > 2, emergency surgery, preoperative morbidities of HTN, DM, heart disease, stroke, cancer, dyslipidemia, Parkinson’s disease, dementia, depression, kidney disease, and liver disease, as well as insomnia and sleep disorder, in addition to alcohol intake and smoking, and medications taken before surgery (calcium channel blockers, diuretics, beta blockers, ACE inhibitors, angiotensin receptor blockers, other antihypertensives, miscellaneous CV drugs, anti-depressants, hypnotics, anti-psychotics, opioids, corticosteroids, antihistamines, and H2 blockers). Only preoperative conditions and the administered drugs were used for matching, and other variables during and after surgery were additionally included in the odds ratio calculation. Data before matching were used for the odds ratio calculation and the ROC curve analysis. We have revised the text in Methods, as follows:
We used 1:2 propensity score matching without replacement. The estimated algorithm was logistic regression and the matching algorithm was the nearest neighbor. The covariates for matching included age, sex, obesity, ASA physical status >2, emergency surgery, preoperative comorbidities of HTN, DM, heart disease, stroke, cancer, dyslipidemia, Parkinson’s disease, dementia, depression, kidney disease, liver disease, insomnia, and sleep disorder, as well as alcohol intake and smoking, in addition to medications taken before surgery (calcium channel blockers, diuretics, beta blockers, ACE inhibitors, angiotensin, receptor blockers, other antihypertensives, miscellaneous CV drugs, anti-depressants, hypnotics, anti-psychotics, opioids, corticosteroids, antihistamines, H2 blockers). Only preoperative conditions and the administrated drugs were used for matching, and other variables during and after surgery were additionally included in the odds ratio calculation.
(page 3, lines 144 – page 4, lines 155)
o Statistical analysis: How exactly were optimal cutpoints determined? Of course we have a ROC displaying all the potential cutpoints, but there are different approaches on which is then the optimal one.
We used the Youden index, as a frequently used summary measure of the receiver operating characteristic curve, to measure the cutoff points for the preoperative inflammatory markers. We have revised the text in Methods, as follows:
The Youden index was used to calculate the cutoff values of the inflammatory markers.
(page 4, lines 163 – page 4, lines 164)
We have address all of the issues raised by the reviewers. We are grateful for the constructive comments that arose during the review process. We believe that our paper has been improved based on these suggestions.
Yours faithfully,
Jong-Hee Sohn, M.D. Ph.D.
Department of Neurology, Chuncheon Sacred Heart Hospital, Hallym University College of Medicine, 77 Sakju-ro, Chuncheon-si, Gangwon-do, 24253, Republic of Korea
Tel: +82-33-252-9970, Fax: +82-33-241-8063
E-mail: deepfoci@hallym.or.kr

Reviewer 2 Report
Comments on the manuscript entitled “Preoperative Inflammatory Markers and the Risk of Postoperative Delirium in Patients Undergoing Lumbar Spinal Fusion Surgery”
Summary
This retrospective study revealed that incidence of postoperative delirium after spine fusion surgery was 7.5% and preoperative inflammation was associated with postoperative delirium. The results are very interesting, but several concerns should be address for publication. In particularly, statistical approach is too poor to provide useful information to readers.
Major comment
Introduction
Neuroinflammation and oxidative stress are listed as causes of delirium, and much has been written about them. However, the authors did not evaluate them and suggested alternatives, such as white blood cell counts, which are available in general hospitals; the relationship between WBC and CRP and delirium should be briefly introduced and why it was limited to spinal surgery should be stated. It is difficult to believe that what is presented in general surgery is not applicable in spine surgery.
Materials and Methods
Including many patients enhances accuracy, but the recruitment duration is too long, a decade. This long recruitment period would changes surgical techniques and anesthesia managements, and their impact cannot be overlooked.
Delirium was defined as an acute condition occurring within 30 days after surgery. Patients should be observed for at least 5 days postoperatively(Br J Anaesth . 2020 Jul;125(1):4-6.). The evaluation period is too short.
Patients undergoing emergency surgery are included; is it possible to draw blood one week in advance in such patients?
Binarization of continuous variables reduces the amount of information. Unless there is a special reason, it should be treated as a continuous variable.
Please submit the definition of each item as a supplemental table.
This is an observational study of the effects on delirium, and it does not make sense to calculate a propensity score with delirium as the outcome.
Calculate the propensity score with some intervention possible or difficult to assign as the objective variable. Factors that arise after the objective variable for which the propensity score is calculated (e.g., operative time, postoperative analgesia, etc.) should then be adjusted for along with the factor of interest in a multivariate analysis using generalized estimating equations, etc.
Results
ASD for age and antidepressants exceeds 0.1.
Inflammation markers are divided into four parts and analyzed or ROC curves are drawn but not described in the method. It may be for a post-hoc analysis, but since it is not a pre-prepared analysis, it lacks the validity of the study, at least since it is not described in the methods.
The discussion cannot be fully evaluated because of inadequate intro, methods, and results.
Author Response
June 8, 2022
Reviewer 2
Journal of Clinical Medicine
Dear Reviewer 2,
Please find attached a revised version of our manuscript, “Preoperative Inflammatory Markers and the Risk of Postoperative Delirium in Patients Undergoing Lumbar Spinal Fusion Surgery” (jcm- 1753872).
We thank you for your thoughtful suggestions regarding the original version of our paper; most of the suggested changes have been incorporated into the revision.
All revisions are described in detail in the order mentioned in the review, following the reviewer’s critique in italics. We believe that the revisions have greatly improved the manuscript and hereby submit the revised version for your consideration for publication.
Comments to author:
This retrospective study revealed that incidence of postoperative delirium after spine fusion surgery was 7.5% and preoperative inflammation was associated with postoperative delirium. The results are very interesting, but several concerns should be address for publication. In particularly, statistical approach is too poor to provide useful information to readers.
We thank the reviewer for these comments and specific suggestions, which have improved our manuscript.
Major comment
Introduction
Neuroinflammation and oxidative stress are listed as causes of delirium, and much has been written about them. However, the authors did not evaluate them and suggested alternatives, such as white blood cell counts, which are available in general hospitals; the relationship between WBC and CRP and delirium should be briefly introduced and why it was limited to spinal surgery should be stated. It is difficult to believe that what is presented in general surgery is not applicable in spine surgery.
In a meta-analysis based on prospective cohort studies, no significant difference in preoperative CRP was observed between people who subsequently developed POD compared with those who did not. Additionally, an increase in preoperative CRP was not associated with POD in the subgroup analysis, which only included studies on participants who did not have preoperative dementia. Serum inflammatory markers based on the differential WBC count, NLR, and PLR are better predictors of mortality and outcome in various medical conditions than are traditional infection markers, including CRP or the total leukocyte count. Additionally, the standardized mean difference of the preoperative inflammatory markers, such as CRP and IL-6 levels, were significantly positive correlated with surgery type in the meta-analysis. Another meta-analysis reported that the associations between preoperative blood levels of inflammatory mediators and POD may be affected by the type of surgery. It is likely that there is a potential contribution of special circumstances associated with certain types of surgery to the etiology of POD. Thus, we analyzed the relevant POD factors by limiting the analysis to one surgery type. We have added text to the Introduction as follows:
These peripheral inflammatory markers are based on the differential WBC count, and are better predictors of mortality and clinical outcome in various medical conditions, than are traditional infection markers, including CRP or the total leukocyte count [13,14]. Additionally, preoperative CRP was not associated with POD in meta-analysis based prospective cohort studies or subgroup analyses, which included only studies on participants who did not have preoperative dementia
(page 2, lines 59 – page 2, lines 64)
The standardized mean difference of preoperative inflammatory markers, such as CRP and IL-6 levels, is significantly positive correlated with surgery type. Another study reported that the association between preoperative blood levels of inflammatory mediators and POD may be affected by the type of surgery. The special circumstances associated with certain types of surgery may contribute to the etiology of POD.
(page 2, lines 67 – page 2, lines 72)
We have also added these citations.
- De Jager C.P.C.,; Wever P.C.,; Gemen E.F.A.,; Kusters R.,; Van Gageldonk-Lafeber A.B.,; Van Der Poll T.,; Laheij R.J.F. The neutrophil-lymphocyte count ratio in patients with community-acquired pneumonia. PLoS ONE. 2012;7:e46561. doi: 10.1371/journal.pone.0046561.
- Núñez J.,; Nunez E.,; Bodí V.,; Sanchis J.,; Minana G.,; Mainar L.,; Santas E.,; Merlos P.,; Rumiz E.,; Darmofal H., et al. Usefulness of the neutrophil to lymphocyte ratio in predicting long-term mortality in ST segment elevation myocardial infarction. Am. J. Cardiol. 2008;101:747–752. doi: 10.1016/j.amjcard.2007.11.004.
Materials and Methods
Including many patients enhances accuracy, but the recruitment duration is too long, a decade. This long recruitment period would changes surgical techniques and anesthesia managements, and their impact cannot be overlooked.
We agree. POD has multiple causes and is affected by various preoperative, intraoperative, and postoperative risk factors. Therefore, we collected limited data, such as one type of surgery to control the variables, and collected data for 11 years. Changes in surgical technique and anesthesia management during the long recruitment period may have affected the results. So, we have added this issue to the limitation section of the Discussion as follows.
We analyzed 11 years of data from the Smart CDW. The long recruitment period allowed for changes in surgical techniques and anesthesia management, and their impact cannot be overlooked.
(page 11, lines 313 – page 11, lines 315)
Delirium was defined as an acute condition occurring within 30 days after surgery. Patients should be observed for at least 5 days postoperatively (Br J Anaesth . 2020 Jul;125(1):4-6.). The evaluation period is too short.
We agree. In our retrospective study, POD patients were defined as those who received postoperative psychiatric counseling and had specific symptoms and signs of delirium, according to the consultation notes. POD was assessed daily by a nurse using the short-form Korean Nursing Delirium Screening Scale. When a nurse suspected POD, the patient received psychiatric counseling by a psychiatrist and the POD diagnosis was confirmed. We created a list of words indicating specific symptoms, based on the electronic medical records. We collected the medical records, including nursing notes and request notes for consultation to a psychiatrist within 48 hours after surgery (1–3 days postoperatively). The psychiatrist responded to the request for consultation within 3–5 days after surgery. When the psychiatrist responded to the request for consultation, POD was confirmed, and the subject was included in the POD group. However, it was difficult to evaluate overall POD due to the short evaluation period. Thus, we have revised the text in the Methods and added this issue to the limitation section of the Discussion as follows.
We collected the medical records, including nursing notes and request notes for consultation with a psychiatrist within 48 hours after surgery (1–3 days postoperatively). When the psychiatrist responded to the request for consultation, POD was confirmed, and the participant was included in the POD group.
(page 3, lines 104 – page 3, lines 108)
Additionally, we collected the medical records, including nursing notes and request notes for consultation to a psychiatrist within 48 hours after surgery. When the psychiatrist responded to the consultation request, POD was confirmed and the patient was included in the POD group. It was difficult to evaluate overall POD due to the short evaluation period.
(page 11, lines 324 – page 12, lines 328)
Patients undergoing emergency surgery are included; is it possible to draw blood one week in advance in such patients?
The description of the research method was incorrect. As a retrospectively designed study, we collected blood test results within 1 week before the surgery. Thus, we have revised the following text in the Methods:
We collected the results of preoperative peripheral blood tests within 1 week before surgery to investigate preoperative inflammatory markers, including the complete blood cell count, and CRP and albumin levels.
(page 3, lines 110 – page 3, lines 113)
Binarization of continuous variables reduces the amount of information. Unless there is a special reason, it should be treated as a continuous variable.
We have reanalyzed and changed the binarized continuous variables to continuous variables. (e.g., age). As recommended, we have revised Table 1, as follows:
Table 1. Preoperative, intraoperative, and postoperative risk factors of the non-POD and POD groups before PSM and after PSM.
Variable |
Before PSM |
After PSM |
||||||||||||
non-POD (n=2,331) |
POD (n=187) |
ASD |
non-POD (n=374) |
POD (n=187) |
ASD |
|||||||||
Age, n (median, IQR) |
65 (56, 72) |
74 (67, 78) |
0.77 |
73 (67, 78) |
74 (67, 78) |
0.05 |
||||||||
Male, n (%) |
1007 (43.4) |
96 (51.3) |
0.16 |
178 (47.6) |
96 (51.3) |
0.07 |
||||||||
Obesity (BMI>29.9), n (%) |
195 (8.4) |
9 (4.8) |
0.17 |
12 (3.2) |
9 (4.8) |
0.07 |
||||||||
ASA physical status>2, n (%) |
714 (30.7) |
112 (59.9) |
0.59 |
235 (62.8) |
112 (59.9) |
0.06 |
||||||||
Emergency surgery, n (%) |
126 (5.4) |
7 (3.7) |
0.09 |
14 (3.7) |
7 (3.7) |
0.00 |
||||||||
HTN, n (%) |
1194 (51.4) |
106 (56.7) |
0.11 |
213 (57.0) |
106 (56.7) |
0.01 |
||||||||
DM, n (%) |
569 (24.5) |
56 (29.9) |
0.12 |
112 (29.9) |
56 (29.9) |
0.00 |
||||||||
Heart disease, n (%) |
267 (11.5) |
35 (18.7) |
0.18 |
64 (17.1) |
35 (18.7) |
0.04 |
||||||||
Stroke, n (%) |
135 (5.8) |
20 (10.7) |
0.16 |
40 (10.7) |
20 (10.7) |
0.00 |
||||||||
Cancer, n (%) |
181 (7.8) |
13 (7.0) |
0.03 |
29 (7.8) |
13 (7.0) |
0.03 |
||||||||
Dyslipidemia, n (%) |
428 (18.4) |
21 (11.2) |
0.23 |
47 (12.6) |
21 (11.2) |
0.04 |
||||||||
Parkinson’s disease, n (%) |
19 (0.8) |
9 (4.8) |
0.19 |
15 (4.0) |
9 (4.8) |
0.04 |
||||||||
Dementia, n (%) |
21 (0.9) |
7 (3.7) |
0.15 |
14 (3.7) |
7 (3.7) |
0.00 |
||||||||
Depression, n (%) |
60 (2.6) |
8 (4.3) |
0.08 |
15 (4.0) |
8 (4.3) |
0.01 |
||||||||
Kidney disease, n (%) |
106 (4.6) |
11 (5.9) |
0.06 |
23 (6.1) |
11 (5.9) |
0.01 |
||||||||
Liver disease, n (%) |
79 (3.4) |
5 (2.7) |
0.05 |
10 (2.7) |
5 (2.7) |
0.00 |
||||||||
Insomnia, n (%) |
143 (6.2) |
17 (9.1) |
0.10 |
36 (9.6) |
17 (9.1) |
0.02 |
||||||||
Sleep disorder, n (%) |
136 (5.9) |
16 (8.6) |
0.10 |
35 (9.4) |
16 (8.6) |
0.03 |
||||||||
Alcohol, n (%) |
583 (25.1) |
32 (17.1) |
0.21 |
57 (15.2) |
32 (17.1) |
0.05 |
||||||||
Smoking, n (%) |
383 (16.5) |
32 (17.1) |
0.02 |
55 (14.7) |
32 (17.1) |
0.06 |
||||||||
Preoperative used drugs |
||||||||||||||
Calcium channel blockers, n (%) |
1011 (43.5) |
81 (43.3) |
0.00 |
156 (41.7) |
81 (43.3) |
0.03 |
||||||||
Diuretics, n (%) |
230 (9.9) |
22 (11.8) |
0.06 |
37 (9.9) |
22 (11.8) |
0.06 |
||||||||
Beta blockers, n (%) |
190 (8.2) |
13 (7.0) |
0.05 |
22 (5.9) |
13 (7.0) |
0.04 |
||||||||
ACE inhibitors, n (%) |
10 (0.4) |
1 (0.5) |
0.01 |
3 (0.8) |
1 (0.5) |
0.04 |
||||||||
Angiotensin receptor blockers, n (%) |
139 (6.0) |
11 (5.9) |
0.00 |
23 (6.1) |
11 (5.9) |
0.01 |
||||||||
Other antihypertensives, n (%) |
29 (1.2) |
1 (0.5) |
0.10 |
1 (0.3) |
1 (0.5) |
0.04 |
||||||||
Miscellaneous CV drugs, n (%) |
126 (5.4) |
13 (7.0) |
0.06 |
25 (6.7) |
13 (7.0) |
0.01 |
||||||||
Anti-depressants, n (%) |
77 (3.3) |
4 (2.1) |
0.08 |
8 (2.1) |
4 (2.1) |
0.00 |
||||||||
Hypnotics, n (%) |
718 (30.9) |
67 (35.8) |
0.10 |
128 (34.2) |
67 (35.8) |
0.03 |
||||||||
Anti-psychotics, n (%) |
168 (7.2) |
33 (17.6) |
0.27 |
55 (14.7) |
33 (17.6) |
0.08 |
||||||||
Opioids, n (%) |
2308 (99.4) |
186 (99.5) |
0.01 |
371 (99.2) |
186 (99.5) |
0.04 |
||||||||
Corticosteroids, n (%) |
644 (27.7) |
68 (36.4) |
0.18 |
145 (38.8) |
68 (36.4) |
0.05 |
||||||||
Antihistamines, n (%) |
815 (35.1) |
95 (50.8) |
0.31 |
183 (48.9) |
95 (50.8) |
0.04 |
||||||||
H2 blockers, n (%) |
687 (29.6) |
54 (28.9) |
0.02 |
111 (29.7) |
54 (28.9) |
0.02 |
||||||||
Intra and postoperative factors |
||||||||||||||
Postop. ICU care, n (%) |
339 (14.6) |
90 (48.1) |
0.67 |
78 (20.9) |
90 (48.1) |
0.54 |
||||||||
Patient-controlled analgesia, n (%) |
2,228 (96.0) |
180 (96.3) |
0.02 |
365 (97.6) |
180 (96.3) |
0.07 |
||||||||
Operation time, hour (median, IQR) |
3.6 (2.8, 4.6) |
3.9 (2.9, 5.1) |
0.26 |
3.7 (2.8, 4.7) |
3.9 (2.9, 5.1) |
0.23 |
||||||||
Surgical range, level (median, IQR) |
2 (1, 2) |
2 (1, 3) |
0.23 |
2 (1, 2) |
2 (1, 3) |
0.15 |
||||||||
Fluid balance (input–output), mL (median, IQR) |
1.0 (0.6, 1.6) |
1.1 (0.6, 1.8) |
0.15 |
1.1 (0.7, 1.8) |
1.1 (0.6, 1.8) |
0.06 |
||||||||
Please submit the definition of each item as a supplemental table.
We included the definitions of the variables as supplementary tables. We have added the following text to the Methods as follows:
We included the definitions of the variables as a supplementary table (Table S3).
(page 3, lines 136 – page 3, lines 137)
Table S3. Definition of each variable
Variables |
Definition |
Preoperative variables |
|
Age |
Age at surgery |
Obesity (BMI > 29.9) |
Obesity was defined as a body mass index greater than 29.9. |
ASA physical status > 2 |
Preoperative scores for American Society of Anesthesiology physical status classification system greater than 2 |
Emergency surgery |
Surgery performed as an emergency |
HTN |
Patients with hypertension as a preoperative comorbidity Patient group classified by hypertension diagnostic code or recorded on admission notes |
DM |
Patients with DM as a preoperative comorbidity Patient group classified by DM diagnostic code or recorded on admission notes |
Heart disease |
Patients with heart disease as a preoperative comorbidity Patient group classified by heart disease diagnostic code or recorded on admission notes |
Stroke |
Patients with stroke as a preoperative comorbidity Patient group classified by stroke diagnostic code or recorded on admission notes |
Cancer |
Patients with cancer as a preoperative comorbidity Patient group classified by cancer diagnostic code or recorded on admission notes |
Dyslipidemia |
Patients with dyslipidemia as a preoperative comorbidity Patient group classified by dyslipidemia diagnostic code or recorded on admission notes |
Parkinson’s disease |
Patients with Parkinson’s disease as a preoperative comorbidity Patient group classified by Parkinson’s disease diagnostic code or recorded on admission notes |
Dementia |
Patients with dementia as a preoperative comorbidity Patient group classified by dementia diagnostic code or recorded on admission notes |
Depression |
Patients with depression as a preoperative comorbidity Patient group classified by depression diagnostic code or recorded on admission notes |
Kidney disease |
Patients with kidney disease as a preoperative comorbidity Patient group classified by kidney disease diagnostic code or recorded on admission notes |
Liver disease |
Patients with liver disease as a preoperative comorbidity Patient group classified by liver disease diagnostic code or recorded on admission notes |
Insomnia |
Patients with insomnia as a preoperative comorbidity Patient group classified by insomnia diagnostic code or recorded on admission notes |
Sleep disorder |
Patients with sleep disorder as a preoperative comorbidity Patient group classified by sleep disorder diagnostic code or recorded on admission notes |
Alcohol |
Patients with a history of drinking alcohol more than once per week at the time of preoperative assessment |
Smoking |
Current smoker at the time of preoperative assessment |
Intra and postoperative variables |
|
Postop. ICU care |
Patients receiving intensive care in the ICU after surgery |
Patient-controlled analgesia |
Patients using a patient-controlled pain therapy device after surgery |
Operation time |
Total time of the surgery |
Surgical range |
The number of vertebrae involved in the lumbar spinal fusion surgery |
Fluid balance |
Fluid volume input minus output during surgery |
This is an observational study of the effects on delirium, and it does not make sense to calculate a propensity score with delirium as the outcome.
The reason we used propensity score matching was to see whether there were any differences in the NMR, MLR, PLR, and CAR in patients who developed POD and those who did not develop POD when other variables before surgery were controlled. Only preoperative conditions and the administered drugs were used for matching, and other variables during and after surgery were additionally included in the odds ratio calculation. We performed propensity score matching and statistical analysis with modifications. We have revised the text in the Methods as follows:
Only preoperative conditions and the administrated drugs were used for matching, and other variables during and after surgery were additionally included in the odds ratio calculation.
(page 4, lines 153 – page 4, lines 155)
Data before matching were used to calculate the odds ratios and to conduct the receiver operating characteristic (ROC) curve analysis.
(page 4, lines 160 – page 4, lines 161)
Also, we have revised Tables 2 and 3, as follows:
Table 2. Differences in the preoperative inflammatory markers between the non-POD and POD groups before and after PSM.
|
Before PSM |
P value |
After PSM |
P value |
|||
non-POD (n=2,322) |
POD (n=187) |
non-POD (n=374) |
POD (n=187) |
||||
NLR median (IQR) |
2.17 (1.55, 3.29) |
2.68 (1.90, 4.19) |
<0.001 |
2.26 (1.72, 3.36) |
2.68 (1.90, 4.19) |
0.002 |
|
MLR median (IQR) |
0.23 (0.17, 0.33) |
0.29 (0.21, 0.39) |
<0.001 |
0.26 (0.19, 0.36) |
0.29 (0.21, 0.39) |
0.049 |
|
PLR median (IQR) |
137.86 (105.02, 182.93) |
140.98 (103.86, 185.35) |
0.464 |
136.94 (104.89, 184.95) |
140.98 (103.86, 185.35) |
0.713 |
|
CAR median (IQR) |
0.26 (0.22, 0.68) |
0.47 (0.25, 1.89) |
<0.001 |
0.27 (0.24, 0.98) |
0.47 (0.25, 1.89) |
0.001 |
|
Table 3. Logistic regression analysis showing the effect of inflammatory markers on developing POD.
|
IQR of inflammatory markers |
|||
|
NLR Q1 |
NLR Q2 |
NLR Q3 |
NLR Q4 |
aOR |
reference |
2.28 |
2.48 |
2.88 |
95% CI |
1.25-4.16 |
1.3-4.73 |
1.39-5.96 |
|
p-value |
0.008 |
0.006 |
0.005 |
|
|
MLR Q1 |
MLR Q2 |
MLR Q3 |
MLR Q4 |
aOR |
reference |
0.87 |
1.07 |
1.47 |
95% CI |
0.48-1.57 |
0.6-1.92 |
0.78-2.76 |
|
p-value |
0.651 |
0.809 |
0.236 |
|
|
PLR Q1 |
PLR Q2 |
PLR Q3 |
PLR Q4 |
aOR |
reference |
0.5 |
0.7 |
0.48 |
95% CI |
0.3-0.84 |
0.42-1.16 |
0.27-0.85 |
|
p-value |
0.008 |
0.169 |
0.011 |
|
|
CAR Q1 |
CAR Q2 |
CAR Q3 |
CAR Q4 |
aOR |
reference |
0.92 |
1.55 |
1.56 |
95% CI |
0.52-1.63 |
0.92-2.63 |
0.91-2.66 |
|
p-value |
0.775 |
0.101 |
0.103 |
Calculate the propensity score with some intervention possible or difficult to assign as the objective variable. Factors that arise after the objective variable for which the propensity score is calculated (e.g., operative time, postoperative analgesia, etc.) should then be adjusted for along with the factor of interest in a multivariate analysis using generalized estimating equations, etc.
We used 1:2 propensity score matching without replacement. The estimation algorithm was logistic regression and the matching algorithm was the nearest neighbor. The covariates for matching included age, sex, obesity, ASA physical status > 2, emergency surgery, preoperative comorbidities of HTN, DM, heart disease, stroke, cancer, dyslipidemia, Parkinson’s disease, dementia, depression, kidney disease, and liver disease, as well as insomnia and sleep disorder, in addition to alcohol intake and smoking, and medications taken before surgery (calcium channel blockers, diuretics, beta blockers, ACE inhibitors, angiotensin receptor blockers, other antihypertensives, miscellaneous CV drugs, anti-depressants, hypnotics, anti-psychotics, opioids, corticosteroids, antihistamines, and H2 blockers). Only preoperative conditions and the administered drugs were used for matching, and the other variables during and after surgery were additionally included in the odds ratio calculation. Data before matching were used for the odds ratio calculation and receiver operating characteristic (ROC) curve analysis. We have revised the text in the Methods as follows:
We used 1:2 propensity score matching without replacement. The estimated algorithm was logistic regression and the matching algorithm was the nearest neighbor. The covariates for matching included age, sex, obesity, ASA physical status >2, emergency surgery, preoperative comorbidities of HTN, DM, heart disease, stroke, cancer, dyslipidemia, Parkinson’s disease, dementia, depression, kidney disease, liver disease, insomnia, and sleep disorder, as well as alcohol intake and smoking, in addition to medications taken before surgery (calcium channel blockers, diuretics, beta blockers, ACE inhibitors, angiotensin, receptor blockers, other antihypertensives, miscellaneous CV drugs, anti-depressants, hypnotics, anti-psychotics, opioids, corticosteroids, antihistamines, H2 blockers).
(page 3, lines 144 – page 4, lines 153)
Results
ASD for age and antidepressants exceeds 0.1.
We have changed the propensity score matching to 1:2, and the ASDs of all variables were < 0.1 after matching. We have revised Table 1.
Inflammation markers are divided into four parts and analyzed or ROC curves are drawn but not described in the method. It may be for a post-hoc analysis, but since it is not a pre-prepared analysis, it lacks the validity of the study, at least since it is not described in the methods.
We have added text to the Methods as follows:
Inflammatory markers were categorized into four groups by quartiles. Multiple logistic regression analysis was performed to evaluate the association between the quartiles of the inflammatory markers and developing POD after adjusting for the variables.
(page 4, lines 157 – page 4, lines 160)
The discussion cannot be fully evaluated because of inadequate intro, methods, and results.
We have re-analyzed the data according to the reviewer’s recommendations, and the Discussion section was revised along with the Introduction, Experimental methods, and Results.
We have addressed all of the issues raised by the reviewers. We are grateful for the constructive comments that arose during the review process. We believe that our paper has been improved based on these suggestions.
Yours faithfully,
Jong-Hee Sohn, M.D. Ph.D.
Department of Neurology, Chuncheon Sacred Heart Hospital, Hallym University College of Medicine, 77 Sakju-ro, Chuncheon-si, Gangwon-do, 24253, Republic of Korea
Tel: +82-33-252-9970, Fax: +82-33-241-8063
E-mail: deepfoci@hallym.or.kr

Round 2
Reviewer 1 Report
Thank you for the revised version of the manuscript. I think most of my points were addressed.
However, I still believe the conclusions regarding the discriminatory ability of the inflammatory markers is rather over-optimistic regarding the relatviely low AUC values. I guess the authors can soften their message here a little bit here.
Author Response
June 17, 2022
Reviewer 1
Journal of Clinical Medicine
Dear Reviewer 1,
Please find attached a revised version of our manuscript, “Preoperative Inflammatory Markers and the Risk of Postoperative Delirium in Patients Undergoing Lumbar Spinal Fusion Surgery” (jcm- 1753872).
We thank you for your suggestions regarding the original version of our paper; most of the suggested changes have been incorporated into the revision.
All of the revisions are described in detail in the order mentioned in the review, following the reviewer’s comments in italics. We believe that the revisions have greatly improved the manuscript and hereby submit the revised version for consideration for publication.
Comments to author:
Thank you for the revised version of the manuscript. I think most of my points were addressed.
We thank the reviewer for these comments and specific suggestions, which have improved our manuscript.
However, I still believe the conclusions regarding the discriminatory ability of the inflammatory markers is rather over-optimistic regarding the relatviely low AUC values. I guess the authors can soften their message here a little bit here.
We agree. We have revised the Abstract, Methods, and Discussion as follows:
Receiver operating characteristic curve analysis showed that the discriminative ability of the NLR, MLR, and CAR for predicting POD was low, but almost acceptable (AUC[95% CI]: 0.60[0.56–0.64], 0.61[0.57–0.65] and 0.63[0.59–0.67], respectively, p<0.001).
(page 1, lines 27 – page 1, lines 29)
The ROC curve analysis showed that the discriminative ability of the NLR, MLR, and CAR for predicting POD was low, but almost acceptable
(page 4, lines 12 – page 4, lines 193)
The ROC curve revealed that the discriminative ability of the NLR, MLR, and CAR for predicting POD development was low but almost acceptable.
(page 11, lines 237 – page 11, lines 238)
We have addressed all of the issues raised by the reviewers. We are grateful for the constructive comments that arose during the review process. We believe that our paper has been improved by these suggestions.
Yours faithfully,
Jong-Hee Sohn, M.D. Ph.D.
Department of Neurology, Chuncheon Sacred Heart Hospital, Hallym University College of Medicine, 77 Sakju-ro, Chuncheon-si, Gangwon-do, 24253, Republic of Korea
Tel: +82-33-252-9970, Fax: +82-33-241-8063
E-mail: deepfoci@hallym.or.kr

Reviewer 2 Report
The comment on the manuscript entitled “Preoperative Inflammatory Markers and the Risk of 2 Postoperative Delirium in Patients Undergoing Lumbar Spinal 3 Fusion Surgery”
Thank you for your revision, but there are some concerns for further proceed.
Introduction
The authors add some explanation, but total balance was not considered. As this reviewer pointed out, the description of inflammatory markers was added in Introduction, which resulted in redundant expression. Moreover, additional description is not study background, but the results of previous study.
Please explain why the authors focused on familiar inflammatory markers in patients undergoing spine surgery and its association with neuroinflammation and oxidative stress.
Methods
Definition of POD
The authors wrote that POD was diagnosed by psychiatrist, but its incidence of 7% in this study higher that previous study. (Many previous study used screening tool for delirium, thus the incidence is not compared simply to this study result) Can you provide the estimated incidence by nurse?
Additionally, 48 hours shows 2 days, not 3 days.
It is conceived that patients undergoing emergency surgery have high inflammatory status, which influence the occurrence of POD. How do the authors think? This association was not considered in this manuscript. Propensity score only adjusted any covariates between patients with and without POD.
Covariates such as surgical time and fluid balance remain to be include for calculation of propensity score.
Statistical approach
The authors changed statistical approach on primary outcome (1:1 to 1:2 matching). This was not planned before; thus, it is not reasonable revision. Statistical methods should be planned prior to analysis. In case the authors analyzed using 1:2 matching, which was not planned before, I recommend that the authors submit this revised version as other new study after approval by Institutional Review Board.
To assess the impacts of some inflammatory markers on POD, propensity score matching seems not to be correct because POD is not modifiable factor. Multiple regression analysis would be a reasonable method.
Author Response
June 17, 2022
Reviewer 2
Journal of Clinical Medicine
Dear Reviewer 2,
Please find attached a revised version of our manuscript, “Preoperative Inflammatory Markers and the Risk of Postoperative Delirium in Patients Undergoing Lumbar Spinal Fusion Surgery” (jcm- 1753872).
We thank you for your suggestions regarding the original version of our paper; most of the suggested changes have been incorporated into the revision.
All of the revisions are described in detail in the order mentioned in the review, following the reviewer’s comments in italics. We believe that the revisions have greatly improved the manuscript and hereby submit the revised version for consideration for publication.
Comments to author:
The comment on the manuscript entitled “Preoperative Inflammatory Markers and the Risk of Postoperative Delirium in Patients Undergoing Lumbar Spinal Fusion Surgery”. Thank you for your revision, but there are some concerns for further proceed.
We thank the reviewer for these comments and suggestions, which helped us to improve our manuscript.
Introduction
The authors add some explanation, but total balance was not considered. As this reviewer pointed out, the description of inflammatory markers was added in Introduction, which resulted in redundant expression. Moreover, additional description is not study background, but the results of previous study.
Please explain why the authors focused on familiar inflammatory markers in patients undergoing spine surgery and its association with neuroinflammation and oxidative stress.
We have added to and revised the Introduction, as follows:
Markers of inflammation and oxidative stress are utilized in research and clinical practice to diagnose and monitor inflammation and oxidative stress.
(page 2, lines 48 – page 2, lines 49)
However, their use in clinical practice is limited due to due to cost or a cumbersome diagnostic procedure. The differential white blood cell (WBC) count is routinely obtained in the majority of hospitalized patients, at no additional cost. The neutrophil-to-lymphocyte ratio (NLR) and platelet-to-lymphocyte ratio (PLR) based on the differential WBC count are readily available markers of generalized inflammation, as reported previously. These peripheral inflammatory markers are better predictors of mortality and clinical outcome in various medical conditions, than are traditional infection markers, including CRP or the total leukocyte count.
(page 2, lines 57 – page 2, lines 65)
Other studies have reported that the association between preoperative blood levels of inflammatory mediators and POD may be affected by the type of surgery.
(page 2, lines 70 – page 2, lines 73)
However, no study has investigated whether preoperative peripheral inflammatory markers are associated with the development of POD after spine surgery. The relationship between blood inflammatory markers and POD remains controversial. We hypothesized that inflammation and oxidative stress are important antecedents of POD in patients undergoing spine surgery.
(page 2, lines 77 – page 2, lines 81)
Methods
Definition of POD
The authors wrote that POD was diagnosed by psychiatrist, but its incidence of 7% in this study higher that previous study. (Many previous study used screening tool for delirium, thus the incidence is not compared simply to this study result) Can you provide the estimated incidence by nurse?
Additionally, 48 hours shows 2 days, not 3 days.
We retrospectively collected electronic medical records, including nursing notes and requests for consultations with a psychiatrist within 48 hours after surgery. POD was assessed daily by a nurse using the short-form Korean Nursing Delirium Screening Scale, and via the nursing notes. When a nurse suspected POD, the patient received psychiatric counseling from a psychiatrist. When the psychiatrist responded to the request for consultation, the subject was included in the POD group following confirmation of the diagnosis.
Of the enrolled patients, 187 (7.4%) developed POD. Additionally, 342 patients had POD according to the nursing records, and the incidence was estimated by the nurses as 13.7%.
The prevalence of POD after lumbar spinal fusion surgery was 7.4%. The prevalence of POD after spinal surgery ranged from 3.8–40.4% and 0.84–27.6% in two previous meta-analyses. The prevalence of POD in our study was relatively low compared to previous studies. The retrospective assessment of POD may have led to underestimation of its incidence. Additionally, patients with hypoactive delirium or mild symptoms of POD who did not consult a psychiatrist may have been excluded, which could explain the relatively low incidence of POD in our study.
We have revised in the Methods and Results as follows.
We collected the medical records, including nursing notes and request notes for consultation with a psychiatrist within 48 hours after surgery.
(page 3, lines 106 – page 3, lines 108)
Of the enrolled patients, 342 had POD according to nursing records, and 187 cases were confirmed by a psychiatrist. Thus, the incidence of POD in our study was 7.4%
(page 4, lines 173 – page 4, lines 175)
It is conceived that patients undergoing emergency surgery have high inflammatory status, which influence the occurrence of POD. How do the authors think? This association was not considered in this manuscript. Propensity score only adjusted any covariates between patients with and without POD.
Covariates such as surgical time and fluid balance remain to be include for calculation of propensity score.
We have performed propensity score matching and a modified statistical analysis. Intraoperative covariates, such as surgical time and fluid balance, were used for matching. We have revised the text in the Methods and Results as follows:
Preoperative condition and the administered drugs, surgical time, and fluid balance were used for matching, and other variables during and after surgery were included in the odds ratio (OR) calculation.
(page 4, lines 150 – page 4, lines 153)
After PSM, the NLR (2.68 [1.90, 4.19] vs. 2.29 [1.70, 3.42], p = 0.004), MLR (0.29 [0.21, 0.39] vs. 0.26 [0.19, 0.36], p = 0.019), and CAR (0.47 [0.25, 1.89] vs 0.29 [0.24, 1.04], p = 0.004) was significantly higher in the POD than non-POD group (Table 2).
(page 4, lines 184 – page 4, lines 187)
We have revised Tables 1 and 2 as follows:
Table 1. Preoperative, intraoperative, and postoperative risk factors of the non-POD and POD groups before and after PSM.
Variable |
Before PSM |
After PSM |
|||||||||||||
non-POD (n=2,331) |
POD (n=187) |
ASD |
non-POD (n=374) |
POD (n=187) |
ASD |
||||||||||
Age, n (median, IQR) |
65 (56, 72) |
74 (67, 78) |
0.77 |
73 (66, 77) |
74 (67, 78) |
0.01 |
|||||||||
Male, n (%) |
1007 (43.4) |
96 (51.3) |
0.16 |
194 (51.9) |
96 (51.3) |
0.01 |
|||||||||
Obesity (BMI > 29.9), n (%) |
195 (8.4) |
9 (4.8) |
0.17 |
11 (2.9) |
9 (4.8) |
0.09 |
|||||||||
ASA physical status > 2, n (%) |
714 (30.7) |
112 (59.9) |
0.59 |
240 (64.2) |
112 (59.9) |
0.09 |
|||||||||
Emergency surgery, n (%) |
126 (5.4) |
7 (3.7) |
0.09 |
18 (4.8) |
7 (3.7) |
0.06 |
|||||||||
HTN, n (%) |
1194 (51.4) |
106 (56.7) |
0.11 |
213 (57.0) |
106 (56.7) |
0.01 |
|||||||||
DM, n (%) |
569 (24.5) |
56 (29.9) |
0.12 |
115 (30.7) |
56 (29.9) |
0.02 |
|||||||||
Heart disease, n (%) |
267 (11.5) |
35 (18.7) |
0.18 |
65 (17.4) |
35 (18.7) |
0.03 |
|||||||||
Stroke, n (%) |
135 (5.8) |
20 (10.7) |
0.16 |
39 (10.4) |
20 (10.7) |
0.01 |
|||||||||
Cancer, n (%) |
181 (7.8) |
13 (7.0) |
0.03 |
31 (8.3) |
13 (7.0) |
0.05 |
|||||||||
Dyslipidemia, n (%) |
428 (18.4) |
21 (11.2) |
0.23 |
37 (9.9) |
21 (11.2) |
0.04 |
|||||||||
Parkinson’s disease, n (%) |
19 (0.8) |
9 (4.8) |
0.19 |
14 (3.7) |
9 (4.8) |
0.05 |
|||||||||
Dementia, n (%) |
21 (0.9) |
7 (3.7) |
0.15 |
12 (3.2) |
7 (3.7) |
0.03 |
|||||||||
Depression, n (%) |
60 (2.6) |
8 (4.3) |
0.08 |
14 (3.7) |
8 (4.3) |
0.03 |
|||||||||
Kidney disease, n (%) |
106 (4.6) |
11 (5.9) |
0.06 |
18 (4.8) |
11 (5.9) |
0.05 |
|||||||||
Liver disease, n (%) |
79 (3.4) |
5 (2.7) |
0.05 |
11 (2.9) |
5 (2.7) |
0.02 |
|||||||||
Insomnia, n (%) |
143 (6.2) |
17 (9.1) |
0.10 |
30 (8.0) |
17 (9.1) |
0.04 |
|||||||||
Sleep disorder, n (%) |
136 (5.9) |
16 (8.6) |
0.10 |
29 (7.8) |
16 (8.6) |
0.03 |
|||||||||
Alcohol, n (%) |
583 (25.1) |
32 (17.1) |
0.21 |
67 (17.9) |
32 (17.1) |
0.02 |
|||||||||
Smoking, n (%) |
383 (16.5) |
32 (17.1) |
0.02 |
68 (18.2) |
32 (17.1) |
0.03 |
|||||||||
Preoperative used drugs |
|||||||||||||||
Calcium channel blockers, n (%) |
1011 (43.5) |
81 (43.3) |
0.00 |
160 (42.8) |
81 (43.3) |
0.01 |
|||||||||
Diuretics, n (%) |
230 (9.9) |
22 (11.8) |
0.06 |
45 (12.0) |
22 (11.8) |
0.01 |
|||||||||
Beta blockers, n (%) |
190 (8.2) |
13 (7.0) |
0.05 |
21 (5.6) |
13 (7.0) |
0.05 |
|||||||||
ACE inhibitors, n (%) |
10 (0.4) |
1 (0.5) |
0.01 |
1 (0.3) |
1 (0.5) |
0.04 |
|||||||||
Angiotensin receptor blockers, n (%) |
139 (6.0) |
11 (5.9) |
0.00 |
20 (5.3) |
11 (5.9) |
0.02 |
|||||||||
Other antihypertensives, n (%) |
29 (1.2) |
1 (0.5) |
0.10 |
1 (0.3) |
1 (0.5) |
0.04 |
|||||||||
Miscellaneous CV drugs, n (%) |
126 (5.4) |
13 (7.0) |
0.06 |
23 (6.1) |
13 (7.0) |
0.03 |
|||||||||
Anti-depressants, n (%) |
77 (3.3) |
4 (2.1) |
0.08 |
8 (2.1) |
4 (2.1) |
0.00 |
|||||||||
Hypnotics, n (%) |
718 (30.9) |
67 (35.8) |
0.10 |
130 (34.8) |
67 (35.8) |
0.02 |
|||||||||
Anti-psychotics, n (%) |
168 (7.2) |
33 (17.6) |
0.27 |
54 (14.4) |
33 (17.6) |
0.08 |
|||||||||
Opioids, n (%) |
2308 (99.4) |
186 (99.5) |
0.01 |
371 (99.2) |
186 (99.5) |
0.04 |
|||||||||
Corticosteroids, n (%) |
644 (27.7) |
68 (36.4) |
0.18 |
135 (36.1) |
68 (36.4) |
0.01 |
|||||||||
Antihistamines, n (%) |
815 (35.1) |
95 (50.8) |
0.31 |
201 (53.7) |
95 (50.8) |
0.06 |
|||||||||
H2 blockers, n (%) |
687 (29.6) |
54 (28.9) |
0.02 |
96 (25.7) |
54 (28.9) |
0.07 |
|||||||||
Intra and postoperative factors |
|||||||||||||||
Postop. ICU care, n (%) |
339 (14.6) |
90 (48.1) |
0.67 |
90 (24.1) |
90 (48.1) |
0.48 |
|||||||||
Patient-controlled analgesia, n (%) |
2,228 (96.0) |
180 (96.3) |
0.02 |
365 (97.6) |
180 (96.3) |
0.07 |
|||||||||
Operation time, hours (median, IQR) |
3.6 (2.8, 4.6) |
3.9 (2.9, 5.1) |
0.26 |
3.9 (3.1, 5) |
3.9 (2.9, 5.1) |
0.09 |
|||||||||
Surgical range, level (median, IQR) |
2 (1, 2) |
2 (1, 3) |
0.23 |
2 (1, 2) |
2 (1, 3) |
0.19 |
|||||||||
Fluid balance (input–output), mL (median, IQR) |
1.0 (0.6, 1.6) |
1.1 (0.6, 1.8) |
0.15 |
1.2 (0.7, 1.8) |
1.1 (0.6, 1.8) |
0.03
|
|||||||||
Table 2. Differences in preoperative inflammatory markers between the non-POD and POD groups before and after PSM
|
Before PSM |
P value |
After PSM |
P value |
|||
non-POD (n=2,322) |
POD (n=187) |
non-POD (n=374) |
POD (n=187) |
||||
NLR median (IQR) |
2.17 (1.55, 3.29) |
2.68 (1.90, 4.19) |
<0.001 |
2.29 (1.70, 3.42) |
2.68 (1.90, 4.19) |
0.004 |
|
MLR median (IQR) |
0.23 (0.17, 0.33) |
0.29 (0.21, 0.39) |
<0.001 |
0.26 (0.19, 0.36) |
0.29 (0.21, 0.39) |
0.019 |
|
PLR median (IQR) |
137.86 (105.02, 182.93) |
140.98 (103.86, 185.35) |
0.464 |
140.99 (104.16, 189.71) |
140.98 (103.86, 185.35) |
0.792 |
|
CAR median (IQR) |
0.26 (0.22, 0.68) |
0.47 (0.25, 1.89) |
<0.001 |
0.29 (0.24, 1.04) |
0.47 (0.25, 1.89) |
0.004 |
|
Statistical approach
The authors changed statistical approach on primary outcome (1:1 to 1:2 matching). This was not planned before; thus, it is not reasonable revision. Statistical methods should be planned prior to analysis. In case the authors analyzed using 1:2 matching, which was not planned before, I recommend that the authors submit this revised version as other new study after approval by Institutional Review Board.
Raw data can be obtained after IRB approval. After obtaining the raw data, we could determine the optimal propensity matching ratio. The IRB approved the use of use of 1:1 to 1:4 matching (depending on the data) a priori, i.e., when we were planning the research.
To assess the impacts of some inflammatory markers on POD, propensity score matching seems not to be correct because POD is not modifiable factor. Multiple regression analysis would be a reasonable method.
Propensity score matching was used to determine if there were differences between the groups with and without POD. As mentioned by the reviewers, we agree that multiple regression analysis is a useful method to evaluate the effect of inflammatory markers on POD. Therefore, we now provide multiple regression results, including novel variables, in addition to the inflammatory marker data.
We revised the Methods and Results as follows:
We calculated the fully adjusted ORs and 95% confidence intervals for developing POD using multivariate logistic regression to evaluate the association between the quartiles of the inflammatory markers, among other variables, and the development of POD.
(page 4, lines 154 – page 4, lines 158)
The ORs of the inflammatory markers and other variables for developing POD are summarized in Table 3.
(page 4, lines 187 – page 4, lines 188)
We revised Table 3 as follows:
Table 3. Multivariate logistic regression showing the effects of inflammatory markers
and other variables on the likelihood of POD development
Variable |
|
Odds ratio (95% CI) |
p value |
|||
IQR of inflammatory markers |
||||||
IQR NLR Q2 |
2.28 (1.25 - 4.16) |
0.008 |
||||
IQR NLR Q3 |
2.48 (1.3 - 4.73) |
0.006 |
||||
IQR NLR Q4 |
2.88 (1.39 - 5.96) |
0.005 |
||||
IQR MLR Q2 |
0.87 (0.48 - 1.57) |
0.651 |
||||
IQR MLR Q3 |
1.07 (0.6 - 1.92) |
0.809 |
||||
IQR MLR Q4 |
1.47 (0.78 - 2.76) |
0.236 |
||||
IQR PLR Q2 |
0.5 (0.3 - 0.84) |
0.008 |
||||
IQR PLR Q3 |
0.7 (0.42 - 1.16) |
0.169 |
||||
IQR PLR Q4 |
0.48 (0.27 - 0.85) |
0.011 |
||||
IQR CAR Q2 |
0.92 (0.52 - 1.63) |
0.775 |
||||
IQR CAR Q3 |
1.55 (0.92 - 2.63) |
0.101 |
||||
IQR CAR Q4 |
1.56 (0.91 - 2.66) |
0.103 |
||||
Other variables |
||||||
Age |
1.05 (1.03 - 1.07) |
<0.001 |
||||
Male |
1.8 (1.22 - 2.65) |
0.003 |
||||
Obesity (BMI > 29.9) |
0.71 (0.33 - 1.53) |
0.387 |
||||
ASA physical status >2 |
2.07 (1.37 - 3.13) |
<0.001 |
||||
Emergency surgery |
0.92 (0.38 - 2.24) |
0.856 |
||||
HTN |
0.77 (0.53 - 1.12) |
0.171 |
||||
DM |
0.94 (0.64 - 1.39) |
0.769 |
||||
Heart disease |
0.71 (0.44 - 1.15) |
0.169 |
||||
Stroke |
1.05 (0.58 - 1.89) |
0.88 |
||||
Cancer |
0.52 (0.28 - 0.99) |
0.047 |
||||
Dyslipidemia |
0.66 (0.4 - 1.11) |
0.119 |
||||
Parkinson’s disease |
4.44 (1.75 - 11.23) |
0.002 |
||||
Dementia |
3.79 (1.4 - 10.27) |
0.009 |
||||
Depression |
2.64 (1.13 - 6.14) |
0.025 |
||||
Kidney_disease |
0.77 (0.37 - 1.61) |
0.494 |
||||
Liver_disease |
0.48 (0.17 - 1.4) |
0.181 |
||||
Insomnia |
0.9 (0.07 - 12.25) |
0.938 |
||||
Sleep disorder |
1.64 (0.11 - 23.48) |
0.716 |
||||
Alcohol |
0.65 (0.4 - 1.08) |
0.096 |
||||
Smoking |
1.1 (0.66 - 1.85) |
0.715 |
||||
Calcium channel blockers |
0.85 (0.6 - 1.21) |
0.374 |
||||
Diuretics |
0.76 (0.43 - 1.32) |
0.33 |
||||
Beta-blockers |
0.79 (0.4 - 1.55) |
0.491 |
||||
ACE_inhibitors |
0.96 (0.08 - 11.91) |
0.976 |
||||
Angiotension receptor blockers |
1.13 (0.54 - 2.37) |
0.752 |
||||
Other_antihypertensives |
0.25 (0.03 - 2.3) |
0.221 |
||||
Miscellaneous_CV_drugs |
1.5 (0.75 - 3.02) |
0.251 |
||||
Antidepressants |
0.64 (0.21 - 1.93) |
0.426 |
||||
Hypnotics_sedatives |
1.07 (0.71 - 1.61) |
0.746 |
||||
Antipsychotics |
3.53 (2.08 - 5.98) |
<0.001 |
||||
Opioids |
0.69 (0.08 - 6.17) |
0.743 |
||||
Corticosteroids |
1.16 (0.8 - 1.68) |
0.429 |
||||
Antihistamines_antiallergics |
1.41 (0.95 - 2.11) |
0.091 |
||||
H2 receptor antagonist |
1.07 (0.71 - 1.61) |
0.742 |
||||
Operation_time |
1.11 (0.97 - 1.26) |
0.125 |
||||
Surgical range |
1.13 (0.96 - 1.32) |
0.142 |
||||
Fluid balance (input-output) |
0.92 (0.75 - 1.12) |
0.391 |
||||
Postop. ICU care |
3.83 (2.6 - 5.65) |
<0.001 |
||||
Patient-controlled analgesia |
1.13 (0.45 - 2.81) |
0.794 |
||||
We have addressed all of the issues raised by the reviewers. We are grateful for the constructive comments that arose during the review process. We believe that our revisions based on these suggestions have improved the manuscript.
Yours faithfully,
Jong-Hee Sohn, M.D. Ph.D.
Department of Neurology, Chuncheon Sacred Heart Hospital, Hallym University College of Medicine, 77 Sakju-ro, Chuncheon-si, Gangwon-do, 24253, Republic of Korea
Tel: +82-33-252-9970, Fax: +82-33-241-8063
E-mail: deepfoci@hallym.or.kr
